# Optimal Trade-offs between Regret and Estimation in Capacitated Multinomial Logit Bandits

## Abstract

Online decision-making involves a fundamental trade-off between two objectives. The first is *regret minimization*, which aims to maximize cumulative reward; the second is *parameter estimation*, which aims to learn the underlying model for downstream tasks. While this trade-off is well-studied in multi-armed bandits (MAB), it remains far less understood in multinomial logit (MNL) bandits, where the decision space is combinatorially large. The only prior work Zuo & Qin (2025) is limited to the uncapacitated case and lacks a tight characterization of the dependence on the number of items $N$. In this work, we establish tight trade-off bounds between regret and customer attraction estimation error for capacitated MNL bandits, with a sharp dependence on $N$. To match these bounds, we introduce an algorithm that achieves the optimal trade-off, providing the first complete characterization of *Pareto optimality* in this setting. The lower-bound technique underlying our results is broadly applicable and also strengthens existing results for MAB. Beyond attraction estimation, our analysis further extends to customer preference estimation error, where the same guarantees continue to hold. As a further application, our framework addresses the joint assortment and pricing problem, yielding new insights into the regret-estimation trade-off in broader contexts.

## 1 Introduction

Assortment optimization is a fundamental problem in revenue management with broad applications in modern retail and e-commerce. Formally, the seller selects a subset of items $S \subseteq [N]$ to offer, typically subject to a capacity constraint $|S| \leq K$. A customer then makes a purchase according to a probabilistic choice model, and the seller's objective is to choose an assortment that maximizes expected revenue. To capture such stochastic choice behavior, a variety of customer choice models have been proposed, including the Multinomial Logit (MNL) (Luce, 1959; Train, 2009; Daganzo, 2014), mixed-logit (McFadden & Train, 2000), Markov-chain choice (Blanchet et al., 2016), and numerous other variants (Bertschek & Kaiser, 2004; Alptekinoğlu & Semple, 2016; Asuncion et al., 2007; Train, 2009; Berbeglia et al., 2022). Among these models, the MNL model, where the customer choices are governed by item-specific *attraction parameters* $\mathbf{v} \in \mathbb{R}^N$, is widely used in both academic research and industrial applications. Its analytical tractability and structural properties make it particularly suitable for large-scale assortment optimization problems (Rusmevichientong et al., 2010a; Davis et al., 2013; Avadhanula et al., 2016).

In data-driven assortment optimization, especially under MNL models, much of the existing literature emphasizes decision-making performance, including online regret minimization (Sauré & Zeevi, 2013; Agrawal et al., 2017) and offline policy learning (Dong et al., 2023; Han et al., 2025). However, the fundamental problem of accurately *estimating attraction parameters* remains critical. Precise estimation is essential not only for identifying the optimal assortment, but also for broader strategic decisions in inventory management and marketing. In these contexts, understanding the intrinsic attractiveness of each product is paramount (Simchi-Levi & Wang, 2025).

In general, reducing estimation error comes at the expense of decision performance, as exploration necessarily involves suboptimal actions. In the multi-armed bandit (MAB) setting, this trade-off has been formalized via Pareto optimality (Simchi-Levi & Wang, 2025), and subsequently extended

to contextual (Li et al., 2024; 2025a) and pricing problems (Simchi-Levi & Wang, 2023). Despite these advances, dedicated studies on attraction estimation in assortment optimization—where the action space is combinatorial and the feedback categorical—remain scarce. The only prior work in this area Zuo & Qin (2025) studies the uncapacitated MNL bandit problems, providing a $\widetilde{\mathcal{O}}(N^{3/4})$ algorithm and an $\Omega(1)$ lower bound. However, the dependency on $N$ remains unresolved, and the capacitated setting is left open. This motivates two fundamental questions:

> *(i) What is the optimal trade-off between estimation error and regret in MNL bandits, particularly concerning its dependency on the number of items $N$?*
> *(ii) Can we design efficient algorithms that achieve this optimal trade-off, especially for the capacitated problem?*

In this work, we address these questions by establishing the first tight characterization of the estimation-regret trade-off for capacitated MNL bandits. To quantify this trade-off, we analyze a metric defined as the product of the estimation error and the square root of the regret, for which we establish a minimax lower bound of $\Omega(\sqrt{N})$. We demonstrate that any algorithm achieving this rate for this metric is Pareto-optimal. Subsequently, we propose an efficient algorithm for the capacitated problem that attains a matching $\widetilde{\mathcal{O}}(\sqrt{N})$ rate, thereby proving its ability to navigate the Pareto frontier. Finally, we showcase the versatility of our framework by extending these tight characterizations to the more complex joint assortment and pricing optimization problem, providing the first Pareto-optimality analysis in that setting.

## 1.1 OUR CONTRIBUTIONS

**A tight lower bound on the estimation-regret trade-off.** We establish a minimax lower bound on the trade-off between estimation error $e_T(\widehat{\mathbf{v}}, \mathbf{v})$ and regret $\text{Reg}_T(\pi, \mathbf{v})$ for the $K$-capacitated MNL bandit problem. We prove that for any policy-estimator pair $(\pi, \widehat{\mathbf{v}})$, there exists an MNL instance $\mathbf{v}$ such that $e_T(\widehat{\mathbf{v}}, \mathbf{v})\sqrt{\text{Reg}_T(\pi, \mathbf{v})} = \Omega(\sqrt{N})$. This result captures the optimal dependency on the number of items $N$ and is a key step toward characterizing Pareto optimality. Our lower bound also strengthens prior results. For the $N$-armed bandit case ($K = 1$), it establishes a tight $\Omega(\sqrt{N})$ lower bound, improving upon the $\Omega(1)$ result from Simchi-Levi & Wang (2025) and matching the existing $\widetilde{\mathcal{O}}(\sqrt{N})$ upper bound.

Our proof strategy combines two new ingredients: (i) an active-exploration lower bound for attraction estimation in MNL bandits, and (ii) a reduction argument connecting Pareto-optimality bounds to active-exploration bounds. To the best of our knowledge, both ingredients are novel and of independent interest. Additionally, we note that the $\Omega(1)$ minimax lower bound claimed in prior works (Simchi-Levi & Wang, 2025; Zuo & Qin, 2025) relies on a flawed argument, for which we provide a detailed counterexample in Appendix A.

**An optimal algorithm on the Pareto frontier.** We propose an efficient algorithm for the capacitated problem that achieves the optimal estimation-regret trade-off. The algorithm is tunable via a parameter $\alpha \in [0, 1/2]$, allowing it to trace the Pareto frontier by attaining an estimation error of $e_T(\widehat{\mathbf{v}}, \mathbf{v}) = \widetilde{\mathcal{O}}((N/T)^{\frac{1-\alpha}{2}})$ and a regret of $\text{Reg}_T(\pi, \mathbf{v}) = \widetilde{\mathcal{O}}(N^\alpha T^{1-\alpha})$. This yields the trade-off $e_T(\widehat{\mathbf{v}}, \mathbf{v})\sqrt{\text{Reg}_T(\pi, \mathbf{v})} = \widetilde{\mathcal{O}}(\sqrt{N})$, which matches our lower bound and thus establishes the Pareto optimality of our algorithm. Our guarantees hold for any capacity $1 \leq K \leq N$. For the special case of the uncapacitated problem ($K = N$), our result improves the prior upper bound of $\widetilde{\mathcal{O}}(N^{3/4})$ from Zuo & Qin (2025) to the tight $\widetilde{\mathcal{O}}(\sqrt{N})$ rate. Furthermore, our analysis removes the restrictive assumption that the no-purchase option must have the largest attraction parameter.

**Extensions to Preference Estimation and Pricing Setting.** We demonstrate the generality of our framework through two key extensions. First, we broaden the analysis from customer attraction estimation to the more general task of preference estimation, showing that the same Pareto-optimality guarantees continue to hold. Second, we apply our results to the joint assortment and pricing problem, where we provide the first tight characterization of the regret-estimation trade-off together with the Pareto-optimality analysis.

Table 1: comparison of different works on parameter estimation and regret minimization trade-off

| Reference | Setting | Estimation object | Upper bound | Lower bound |
|---|---|---|---|---|
| Simchi-Levi & Wang (2025) | MAB | Reward | $\widetilde{\mathcal{O}}(\sqrt{N})$ | $\Omega(1)$ |
| Li et al. (2024) | Contextual MAB[†] | Reward | $\mathcal{O}(\sqrt{M})$ | $\Omega(\sqrt{M})$ |
| Simchi-Levi & Wang (2023) | Single-item pricing | Demand parameter | $\mathcal{O}(1)$ | $\Omega(1)$ |
| Zuo & Qin (2025) | MNL bandits | Attraction parameter | $\widetilde{\mathcal{O}}(N^{3/4})$ | $\Omega(1)$ |
| Zuo & Qin (2025) | MNL bandits | Revenue | $\widetilde{\mathcal{O}}(N^{5/2})$ | $\Omega(1)$ |
| *This work* | | | | |
| Theorem 3.2 & 4.2 | MNL bandits | Attraction parameter | $\widetilde{\mathcal{O}}(\sqrt{N})$ | $\Omega(\sqrt{N})$ |
| Remark 4.2 | MNL bandits | Revenue | $\widetilde{\mathcal{O}}(K\sqrt{N})$ | $\Omega(1)$ |
| Theorem 5.1 | MNL bandits | Preference | $\widetilde{\mathcal{O}}(\sqrt{N})$ | $\Omega(\sqrt{N})$ |
| Theorem 5.2 | Joint assortment & pricing optimization | Demand parameter | $\widetilde{\mathcal{O}}(\sqrt{N})$ | $\Omega(\sqrt{N})$ |

[†] Li et al. (2024) considers the setting with $N = 2$ and $M$ different contexts.

## 1.2 RELATED WORKS

**MNL bandits.** Regret minimization in MNL bandits is a well-established area of research (Caro & Gallien, 2007; Rusmevichientong et al., 2010a; Sauré & Zeevi, 2013; Agrawal et al., 2017; 2019; Chen et al., 2019; 2021b; Saha & Gaillard, 2024). For the capacitated problem, a minimax optimal regret of $\widetilde{\Theta}(\sqrt{NT})$ has been established (Agrawal et al., 2017; 2019; Chen & Wang, 2018). In the uncapacitated setting, the optimal regret is $\widetilde{\Theta}(\sqrt{T})$, which is independent of the number of items $N$ (Chen et al., 2019; 2021b). Recent work has also extended the problem to include additional constraints (Cheung & Simchi-Levi, 2017; Aznag et al., 2024; Chen et al., 2024) and more complex choice models (Ou et al., 2019; Oh & Iyengar, 2021; Chen et al., 2021a; Goyal & Perivier, 2022; Zhang & Sugiyama, 2024; Li et al., 2025b; Lee & hwan Oh, 2025). Finally, beyond the MNL bandits, where only the assortment is chosen, there has been growing interest in the more general dynamic joint assortment-pricing problem under MNL models, which captures both product selection and pricing decisions simultaneously (Miao & Chao, 2021; Erginbas et al., 2025; Kim & Oh, 2025).

**Estimation-regret trade-off in bandit problems.** Characterizing the trade-off between pure exploration and regret minimization is an active area of research in multi-armed bandits, often framed through the lens of Pareto optimality. Two main lines of work have emerged. The first focuses on the trade-off between best-arm identification and regret minimization, where the exploration goal is to identify the optimal action (Degenne et al., 2019; Zhong et al., 2023; Qin & Russo, 2024; Kim et al., 2023). The second line of work defines the exploration objective more broadly as learning the underlying reward model for all actions, as explored in MAB and contextual MAB settings (Simchi-Levi & Wang, 2025; Li et al., 2024; 2025a; Simchi-Levi & Wang, 2023). Our work extends this second framework to the MNL bandits, which features a combinatorial action space. The most closely related work is Zuo & Qin (2025), which also studies this trade-off in MNL bandits but is restricted to the uncapacitated case and provides a weaker characterization of the dependency on the number of items $N$.

## 1.3 NOTATIONS

We use $[N]$ to denote the set of items $\{1, \ldots, N\}$. For a fixed vector $\mathbf{v} = (v_1, \cdots, v_N) \in \mathbb{R}^N$, we define $\|\mathbf{v}\|_\infty := \max_{1 \le i \le N} |v_i|$. For functions $f = f(T, N)$ and $g = g(T, N)$, we use standard asymptotic notation $f = \mathcal{O}(g)$ if $f$ is upper-bounded by $g$ up to a constant factor, $f = \Omega(g)$ if $g = \mathcal{O}(f)$, and $f = \Theta(g)$ if both hold. The tilde notations $\widetilde{\mathcal{O}}(\cdot), \widetilde{\Omega}(\cdot), \widetilde{\Theta}(\cdot)$ suppress polylogarithmic factors. We also write $f \lesssim g$ for $f = \widetilde{\mathcal{O}}(g)$, $f \gtrsim g$ for $f = \widetilde{\Omega}(g)$, and $f \asymp g$ for $f = \widetilde{\Theta}(g)$.

## 2 PRELIMINARIES

**MNL choice models.** At each time $t$, the seller offers an assortment $S_t \subseteq \{1, \ldots, N\}$. A customer selects $c_t \in S_t \cup \{0\}$, where 0 represents the no-purchase option. To characterize customer

preferences, we define *attraction vector* $\mathbf{v} = (v_1, \ldots, v_N)$ and let $v_0 = 1$ denote the no-purchase preference. The attraction vector $\mathbf{v}$ is unknown to the seller. Then the probability that a customer purchases product $i$ from assortment $S_t$ is given by

$$\mathbb{P}_{\mathbf{v}}\left(c_t = i \mid S_t\right) = \begin{cases} \frac{v_i}{1 + \sum_{j \in S_t} v_j}, & \text{if } i \in S_t \cup \{0\}, \\ 0, & \text{otherwise.} \end{cases} \tag{1}$$

We define *revenue vector* $\mathbf{r} = (r_1, \ldots, r_N)$ with $0 \leq r_i \leq 1$, where each product $i$ yields a known revenue $r_i$ upon purchase, and let $r_0 = 0$ denote the revenue of no-purchase option. The expected revenue from offering assortment $S_t$ under the attraction vector $\mathbf{v} = (v_1, \ldots, v_N)$ is given by

$$R(S_t, \mathbf{v}) = \sum_{i \in S_t} r_i \mathbb{P}_{\mathbf{v}}\left(c_t = i \mid S_t\right) = \sum_{i \in S_t} \frac{r_i v_i}{1 + \sum_{k \in S_t} v_k}. \tag{2}$$

In the online assortment optimization problem, the seller employs a policy $\pi = (\pi_1, \ldots, \pi_T)$ to sequentially choose assortments $(S_1, \ldots, S_T)$ over a horizon of $T$ periods. The goal is twofold: to minimize regret and to accurately estimate the attraction vector $\mathbf{v}$.[1]

The regret of a policy $\pi$ measures the total expected revenue loss against the optimal static assortment $S^\star = \arg\max_{S \in \mathcal{S}} R(S, \mathbf{v})$:

$$\text{Reg}_T(\pi, \mathbf{v}) = \sum_{t=1}^{T} R(S^\star, \mathbf{v}) - \mathbb{E}[R(S_t, \mathbf{v})].$$

Concurrently, the quality of learning is measured by the estimation error of the attraction vector $\mathbf{v}$. After $T$ periods, given an estimator $\widehat{\mathbf{v}} = (\widehat{v}_1, \ldots, \widehat{v}_N)$, the estimation error is defined as:

$$e_T(\widehat{\mathbf{v}}, \mathbf{v}) = \mathbb{E}[\|\widehat{\mathbf{v}} - \mathbf{v}\|_\infty].$$

Throughout this paper, we impose the following standard assumption:

**Assumption 2.1** (Large time horizon)**.** *The number of products $N$ is small relative to the time horizon $T$; specifically, there exists a universal constant $C_1 \geq 1$ such that $C_1 N \log N \leq T$.*

Assumption 2.1 is a standard technical condition for settings where the time horizon is substantially larger than the number of products to ensure precise estimation.

**Pareto optimality.** We are interested in simultaneously minimizing regret and estimation error. This leads to the following bi-objective optimization problem, where we seek to minimize the worst-case performance over all possible environments:

$$\inf_{(\pi, \widehat{\mathbf{v}})} \sup_{(\mathbf{v}, \mathbf{r}) \in \mathcal{E}} \left(e_T(\widehat{\mathbf{v}}, \mathbf{v}), \text{Reg}_T(\pi, \mathbf{v})\right).$$

Here, a solution is a policy-estimator pair $(\pi, \widehat{\mathbf{v}})$, and its performance is characterized by the vector of its worst-case estimation error and regret. To compare different solutions, we use the concept of Pareto dominance.

**Definition 2.1** (Pareto Dominance)**.** *A solution $(\pi_1, \widehat{\mathbf{v}}_1)$ Pareto dominates another solution $(\pi_2, \widehat{\mathbf{v}}_2)$ if its worst-case performance is no worse in either objective and strictly better in at least one. Formally, this means*

$$\sup_{(\mathbf{v}, \mathbf{r}) \in \mathcal{E}} e_T(\widehat{\mathbf{v}}_1, \mathbf{v}) \lesssim \sup_{(\mathbf{v}, \mathbf{r}) \in \mathcal{E}} e_T(\widehat{\mathbf{v}}_2, \mathbf{v}) \quad and \quad \sup_{(\mathbf{v}, \mathbf{r}) \in \mathcal{E}} Reg_T(\pi_1, \mathbf{v}) \lesssim \sup_{(\mathbf{v}, \mathbf{r}) \in \mathcal{E}} Reg_T(\pi_2, \mathbf{v}),$$

*and at least one inequality holds strictly.*[2]

---

[1] In contrast, prior work by Zuo & Qin (2025) considers the Average Treatment Effect (ATE) estimation error, defined as $e_T(\widehat{\Delta}, \Delta) := \mathbb{E}[\|\widehat{\Delta} - \Delta\|_\infty]$, where $\Delta = [v_i - v_j]_{1 \leq i < j \leq N}$ and $\widehat{\Delta}$ is an estimator of $\Delta$. However, the ATE is tailored for experimental design, where the primary goal is to estimate the relative differences between items (Simchi-Levi & Wang, 2025). This metric is less suitable for the MNL bandits, where the objective is to learn the underlying attraction parameters from preference feedback, not just their pairwise differences. Nevertheless, our results on the estimation error $e_T(\widehat{\mathbf{v}}, \mathbf{v})$ directly imply bounds of the same order for the ATE estimation error $e_T(\widehat{\Delta}, \Delta)$; see details in Appendix F.

[2] i.e., is asymptotically smaller after ignoring polylogarithmic factors.

This leads to the notion of an optimal solution in the Pareto sense.

**Definition 2.2** (Pareto Optimality and Frontier). *A solution* $(\pi^*, \widehat{\mathbf{v}}^*)$ *is* Pareto optimal *if no other feasible solution Pareto dominates it. The set of performance vectors corresponding to all Pareto optimal solutions constitutes the* Pareto frontier.

The Pareto frontier characterizes the fundamental trade-off between regret and estimation error. Any point on this frontier represents a solution where one objective cannot be improved without degrading the other. Our goal is to design policies whose performance lies on this Pareto frontier.

## 3 FUNDAMENTAL LIMITS OF THE REGRET-ESTIMATION TRADE-OFF

To understand the intrinsic difficulty of balancing regret minimization and parameter estimation, we begin by establishing lower bounds. We aim to answer the question: What is the best possible trade-off between regret and estimation error that any algorithm can achieve? To formalize this, we analyze the metric $e_T(\widehat{\mathbf{v}}, \mathbf{v})\sqrt{\mathrm{Reg}_T(\pi, \mathbf{v})}$, which captures the interplay between these two competing objectives.

**The lower bound on estimation error.** As a preliminary step towards our main trade-off bound, we first establish a lower bound on the attraction estimation error $e_T(\widehat{\mathbf{v}}, \mathbf{v})$ in isolation. This result quantifies the statistical difficulty of learning the underlying parameters, irrespective of the regret.

**Theorem 3.1.** *For all policy-estimator pairs* $(\pi, \widehat{\mathbf{v}})$*, there exists a hard instance* $(\mathbf{v}, \mathbf{r}) \in \mathcal{E}$ *such that*

$$e_T(\widehat{\mathbf{v}}, \mathbf{v}) \geq \frac{1}{16}\sqrt{\frac{N}{T}}.$$

Theorem 3.1 shows that any admissible policy-estimator pair must incur estimation error at least on the order of $\Omega(\sqrt{N/T})$. Intuitively, even if the learner is willing to suffer arbitrarily large regret, a certain number of effective samples are required to estimate attraction parameters, and this creates an irreducible floor. This result holds for any capacity constraint $1 \leq K \leq N$.

The proof of Theorem 3.1 introduces a novel construction of hard instances centered on the least-explored item and defined through a quantity that captures information gain. More precisely, we define a weighted count of an item's appearances, $W_i = \sum_{t=1}^{T} \mathbb{1}\{i \in S_t\}/(1+|S_t|)$. The weighting factor $1/(1+|S_t|)$ is crucial, as it naturally captures how the information gained about any single item is diluted by the size of the assortment $|S_t|$. The detailed proof is provided in Appendix B.1.

**The trade-off lower bound.** Building on the estimation lower bound, we now present our main result for this section, which formalizes the fundamental trade-off that any policy must obey.

**Theorem 3.2.** *Suppose* $K \leq N/8$*. For any exploration-exploitation policy* $\pi$ *and attraction estimator* $\widehat{\mathbf{v}}$*, there exists a hard instance* $(\mathbf{v}, \mathbf{r}) \in \mathcal{E}$ *such that*

$$e_T(\widehat{\mathbf{v}}, \mathbf{v})\sqrt{Reg_T(\pi, \mathbf{v})} \geq C\sqrt{N},$$

*for some universal constant* $C > 0$*.*

**Remark 3.1** (On a Flawed Argument in Related Literature). *Theorem 1 in Simchi-Levi & Wang (2025) establishes an* $\Omega(1)$ *lower bound for a similar metric in the MAB setting. However, their proof relies on a flawed intermediate result, for which we provide a counterexample in Appendix A. Consequently, their argument is insufficient to establish their main theorem. This issue also affects subsequent work Zuo & Qin (2025), which uses the same line of reasoning. Our proof is self-contained and provides a correct derivation.*

Theorem 3.2 presents our central lower bound, quantifying the inherent tension between exploration and exploitation. It establishes that any policy aggressively minimizing regret must limit its exploration, thereby incurring a higher worst-case estimation error. Our $\Omega(\sqrt{N})$ bound is the first to establish a tight dependence on the number of items $N$, strengthening prior results in related settings and providing a sharp benchmark for algorithm performance. The capacity constraint $K \leq N/8$ is arises naturally from Chen & Wang (2018) in the regret analysis component of our proof. As discussed in their work, this condition is mild and satisfied in many practical scenarios. The detailed proof is in Appendix B.2.

---

**Algorithm 1 Function** `estimation(N, r, T, K)`

---

1: **Input:** number of products $N$, revenue vector $\mathbf{r} = (r_1, \ldots, r_N) \in [0,1]^N$, time horizon $T$, assortment capacity $K$
2: $t = 1; \ell = 1$ `// keeps track of the time steps and number of epochs`
3: $\mathcal{E}_1 = \varnothing; T_i(1) = 0, i = 1, \ldots, N$
4: **while** $t < T$ **do**
5:    `/* Choose the assortment "evenly" */`
6:    Select $S_\ell \subseteq [N]$ as the set of $K$ products with the fewest offered epochs $T_i(\ell)$
7:    **repeat**
8:      Offer assortment $S_t = S_\ell$, and observe customer choice $c_t \in S_\ell \cup \{0\}$
9:      $\mathcal{E}_\ell \leftarrow \mathcal{E}_\ell \cup t; t \leftarrow t + 1$
10:    **until** $c_t = 0$ `// no-purchase happens`
11:    **for** $i \in S_\ell$ **do**
12:      Compute $\widehat{v}_{i,\ell} = \sum_{t \in \mathcal{E}_\ell} \mathbb{1}\{c_t = i\}$ `// number of selections for product i`
13:      Update $\mathcal{T}_i(\ell) = \{\tau \le \ell \mid i \in S_\tau\}$, $T_i(\ell) = |\mathcal{T}_i(\ell)|$ `// epochs with product i offered`
14:      Update $\widehat{v}_i = \frac{1}{T_i(\ell)} \sum_{\tau \in \mathcal{T}_i(\ell)} \widehat{v}_{i,\tau}$ `// sample mean of the estimates`
15:    **end for**
16:    $\ell \leftarrow \ell + 1; \mathcal{E}_\ell = \varnothing$
17: **end while**
18: **Output:** attraction estimator $\widehat{\mathbf{v}} = (\widehat{v}_1, \ldots, \widehat{v}_N)$, sequence of assortments $S_1, \ldots, S_T \subseteq [N]$

---

**Relation with Pareto optimality.** The minimax lower bound from Theorem 3.2 serves not just as a limit but also as a certificate of optimality. This leads directly to a sufficient condition for an algorithm to be Pareto optimal.

**Theorem 3.3.** *A policy-estimator pair $(\pi, \widehat{\mathbf{v}})$ is Pareto optimal if for all instances $(\mathbf{v}, \mathbf{r}) \in \mathcal{E}$, it achieves*

$$e_T(\widehat{\mathbf{v}}, \mathbf{v}) \sqrt{Reg_T(\pi, \mathbf{v})} = \widetilde{\mathcal{O}}(\sqrt{N}).$$

This theorem provides a clear target for algorithm design. It states that any algorithm achieving a trade-off performance that matches our lower bound (up to polylogarithmic factors) is guaranteed to lie on the Pareto frontier. In the following section, we present an algorithm that meets this condition.

# 4 AN OPTIMAL ALGORITHM ON THE PARETO FRONTIER

Having established the fundamental $\Omega(\sqrt{N})$ limit on the regret-estimation trade-off, we now demonstrate that this limit is achievable. In this section, we present a novel and efficient algorithm for the capacitated MNL bandit problem that matches our lower bound. This constructively proves the tightness of our bound and provides the first algorithm proven to be Pareto optimal for this setting.

Specifically, we consider the $K$-capacitated MNL Bandit problem with $\mathcal{S} := \{S \subseteq [N] : |S| \le K\}$ as the collection of all feasible assortments. Throughout this section, we impose tsumption, which generalizes the assumption of $v_i \in [0,1]$ made in prior work (Zuo & Qin, 2025).

**Assumption 4.1** (Bounded attraction parameters). *The attraction vector $\mathbf{v} = (v_1, \ldots, v_N)$ is bounded, i.e., there exists a constant $V \ge 1$ such that $v_i \in [0, V]$ for all $i \in [N]$.*

## 4.1 AN ALGORITHM FOR PURE ESTIMATION

We begin by analyzing a specialized algorithm designed for the pure-exploration task of minimizing estimation error. This algorithm serves as a crucial building block for our main result and represents one extreme of the Pareto frontier.

Algorithm 1 provides a simple yet effective strategy for parameter estimation. It operates in epochs, where each epoch continues until a no-purchase event is observed. In each epoch $\ell$, the algorithm greedily selects the $K$ items that have been included in the fewest past epochs. This 'round-robin' approach over assortments ensures that all items receive a balanced number of observations over time, which is critical for minimizing the maximum estimation error.

---

**Algorithm 2** Regret-Estimation Error Trade-off for $K$-Capacitated MNL Bandit

---

1: **Input:** number of products $N$, revenue vector $\mathbf{r} = (r_1, \ldots, r_N) \in [0,1]^N$, time horizon $T$, assortment capacity $K$, trade-off parameter $\alpha \in [0, 1/2]$
2: Calculate estimation steps $T_e = \lceil N^\alpha T^{1-\alpha} \rceil$ // Assumption 2.1 ensures $T_e \leq T$
3: $\widehat{v}_1, \ldots, \widehat{v}_N, \quad S_1, \ldots, S_{T_e} = $ estimation$(N, \mathbf{r}, T_e, K)$  // Algorithm 1 for attraction estimation
4: $S_{T_e+1}, \ldots, S_T = $ regret-capacitated$(N, \mathbf{r}, T - T_e, K)$ // Algorithm 3 for regret minimization

---

**Theorem 4.1.** *For any bounded instance $\mathbf{v} = (v_1, \ldots, v_N)$ of the $K$-capacitated MNL Bandit problem with $N$ products and revenues $r_i \in [0,1]$, the estimation error $e_T(\widehat{\mathbf{v}}, \mathbf{v})$ of the admissible pair $(\pi, \widehat{\mathbf{v}})$ produced by Algorithm 1 at time $T$ satisfy*

$$e_T(\widehat{\mathbf{v}}, \mathbf{v}) \leq C_2 V^{3/2} \sqrt{\frac{N \log N}{T}},$$

*where $C_2 > 0$ is a universal constant.*

**Remark 4.1** (Implication for Revenue Estimation). *A direct consequence of Theorem 4.1 is a sharper bound on the uniform revenue estimation error. The vector of expected revenues is $\mathbf{R}(\mathbf{v}) = \{R(S, \mathbf{v})\}_{S \in \mathcal{S}}$, where $R(S, \mathbf{v})$ is defined in (2). Using the plug-in estimator $\widehat{\mathbf{R}} = \mathbf{R}(\widehat{\mathbf{v}})$, the uniform error $e_T(\widehat{\mathbf{R}}, \mathbf{R}(\mathbf{v})) := \mathbb{E}[\|\widehat{\mathbf{R}} - \mathbf{R}(\mathbf{v})\|_\infty]$ is bounded by $\widetilde{\mathcal{O}}(K\sqrt{N/T})$. This substantially improves upon the previous $\mathcal{O}(N^2 \sqrt{N/T})$ bound of Zuo & Qin (2025), illustrating that accurate attraction estimation directly translates into precise revenue predictions. The details are deferred to Appendix C.5.*

Theorem 4.1 demonstrates that our pure-exploration strategy achieves an estimation error of $\widetilde{\mathcal{O}}(\sqrt{N/T})$, matching the rate of our lower bound in Theorem 3.1. Furthermore, since the per-period regret is at most 1 (as $r_i \in [0,1]$), the total regret is bounded by $\text{Reg}_T(\pi, \mathbf{v}) \leq T$. Combining this with the result from Theorem 4.1, we can bound the trade-off metric $e_T(\widehat{\mathbf{v}}, \mathbf{v})\sqrt{\text{Reg}_T(\pi, \mathbf{v})} \leq C_2 V^{3/2}\sqrt{N \log N}$. This matches the lower bound from Theorem 3.1 up to logarithmic factors, proving that this pure estimation algorithm is *Pareto optimal*. It represents the extreme point on the Pareto frontier that prioritizes exploration.

### 4.2 NAVIGATING THE PARETO FRONTIER

While Algorithm 1 is optimal for pure estimation, a practical system must allow for balancing both objectives. To achieve this, we introduce Algorithm 2, a flexible algorithm designed to trace the Pareto frontier.

Algorithm 2 employs a simple and intuitive two-phase structure. It first runs a pure *estimation phase* for $T_e$ time steps using Algorithm 1 to obtain a reliable estimate $\widehat{\mathbf{v}}$. For the remaining $T - T_e$ steps, it switches to a *regret minimization phase*, using a UCB-style policy (Algorithm 3) proposed by Agrawal et al. (2017) (see Appendix C.3 for details). The balance between these phases is controlled by a single, user-specified trade-off parameter $\alpha \in [0, 1/2]$, where $T_e = \lceil N^\alpha T^{1-\alpha} \rceil$. A smaller $\alpha$ favors estimation, while a larger $\alpha$ prioritizes regret minimization.

**Theorem 4.2.** *For any bounded instance $\mathbf{v}$ of the $K$-capacitated MNL Bandit problem and any trade-off parameter $\alpha \in [0, 1/2]$, the admissible pair $(\pi, \widehat{\mathbf{v}})$ produced by Algorithm 2 at time $T$ satisfies the following bounds:*

*(i) The estimation error is bounded by: $e_T(\widehat{\mathbf{v}}, \mathbf{v}) \leq C_2 V^{3/2} \sqrt{\dfrac{N^{1-\alpha} \log N}{T^{1-\alpha}}}$.*

*(ii) The regret is bounded by: $\text{Reg}_T(\pi, \mathbf{v}) \leq C_3 V N^\alpha T^{1-\alpha} \log^2 NT$.*

*(iii) The trade-off metric is bounded by: $e_T(\widehat{\mathbf{v}}, \mathbf{v})\sqrt{\text{Reg}_T(\pi, \mathbf{v})} \leq C_4 V^2 \sqrt{N \log N \log^2 NT}$.*

*Here, $C_2, C_3, C_4 > 0$ are universal constants.*

**Remark 4.2** (Revenue Estimation Trade-off). *Following the same logic as Remark 4.1, the bounds from Theorem 4.2 also imply a sharp bound on the trade-off for revenue estimation. Specifically,*

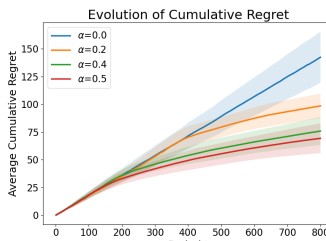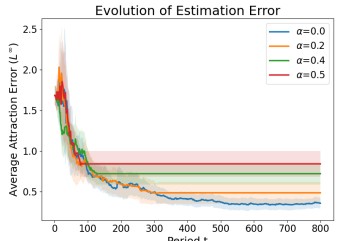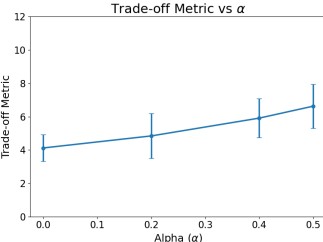

Figure 1: Performance of Algorithm 2 for $N = 8$, $K = 5$, $T = 800$ with varying $\alpha$. Left: Average cumulative regret with 95% CIs. Middle: Average $L^\infty$ estimation error with 95% CIs. Right: Trade-off metric $e_T(\widehat{\mathbf{v}}, \mathbf{v})\sqrt{\text{Reg}_T(\pi, \mathbf{v})}$ vs. $\alpha$. Results are averaged over 20 runs.

*the trade-off metric is bounded by $e_T(\widehat{\mathbf{R}}, \mathbf{R}(\mathbf{v}))\sqrt{Reg_T(\pi, \mathbf{v})} = \widetilde{\mathcal{O}}(K\sqrt{N})$. This significantly improves upon the previous $\widetilde{\mathcal{O}}(N^{5/2})$ bound of Zuo & Qin (2025). See details in Appendix C.5.*

Theorem 4.2 provides our main upper bound and the central constructive result of this paper. By tuning the parameter $\alpha$, our algorithm can trace the Pareto frontier, achieving an estimation error bounded by $\widetilde{\mathcal{O}}((N/T)^{\frac{1-\alpha}{2}})$ and a regret bounded by $\widetilde{\mathcal{O}}(N^\alpha T^{1-\alpha})$. Most importantly, part (iii) demonstrates that our algorithm achieves the trade-off metric of $\widetilde{\mathcal{O}}(\sqrt{N})$. This result matches our minimax lower bound from Theorem 3.2 and, by virtue of Theorem 3.3, formally proves that Algorithm 2 is *Pareto optimal*. Our algorithm generalizes to all capacities $1 \le K \le N$, including the uncapacitated ($K = N$) and Multi-Armed Bandit ($K = 1$) special cases. Compared with Zuo & Qin (2025), our method improves the uncapacitated upper bound from $\widetilde{\mathcal{O}}(N^{3/4})$ to the tight $\widetilde{\mathcal{O}}(\sqrt{N})$. This provides the first complete and tight characterization of the fundamental trade-off between regret and estimation in capacitated MNL bandits. In Appendix G, we introduce an anytime version of Algorithm 2 using the doubling-trick technique, which retains the same performance guarantees without requiring prior knowledge of the time horizon $T$.

### 4.3 NUMERICAL EXPERIMENTS

We conduct numerical experiments to empirically validate the performance of Algorithm 2 and illustrate the trade-off between regret minimization and estimation error. The results confirm that the trade-off parameter $\alpha$ effectively allows navigation of the Pareto frontier. In Appendix H, we present additional experiments comparing Algorithm 2 with the baseline in Zuo & Qin (2025).

**Experimental setup.** In our simulations, we consider a setting with $N = 8$ products and a seller capacity of $K = 5$. The time horizon is set to $T = 800$. For each simulation run, the true attraction parameters $v_i$ are drawn uniformly from $[0, V]$ with $V = 2$, and revenues $r_i$ are drawn from $[0, 1]$. We evaluate Algorithm 2 for different values of the trade-off parameter $\alpha \in \{0.0, 0.2, 0.4, 0.5\}$. The results are averaged over 20 independent runs.

**Results and analysis.** Figure 1 empirically validates our theoretical findings. As predicted, the trade-off parameter $\alpha$ controls the balance between exploration and exploitation. The left panel shows that a smaller $\alpha$ (longer estimation phase $T_e$) leads to higher cumulative regret. Conversely, the middle panel demonstrates that a smaller $\alpha$ results in a lower final estimation error. The right panel shows that the trade-off metric remains nearly constant across different values of $\alpha$, consistent with our theoretical prediction of Pareto optimality. These results confirm that by tuning $\alpha$, Algorithm 2 can effectively trace the Pareto frontier, enabling a practitioner to select a desired balance between minimizing regret and achieving accurate parameter estimation.

### 5 EXTENSIONS

This section demonstrates the broader applicability of our framework by extending the analysis to two related problems: estimating pairwise preferences and the joint assortment-pricing problem. We only present the main results here, with detailed proofs and algorithmic descriptions provided in Appendices D and E.

**Trade-offs with Preference Feedback Estimation.** Preference-based online learning has recently gained significant attention (Bengs et al., 2021). In this context, we aim to estimate pairwise preferences. Let $\mathcal{N}_0 = [N] \cup \{0\}$ be the set of items, including the no-purchase option. For any pair $i, j \in \mathcal{N}_0$, the probability of choosing item $i$ over $j$ is

$$p_{ij} := \mathbb{P}(i \succ j) = \frac{v_i}{v_i + v_j}.$$

We denote the vector of these probabilities by $\mathbf{p}(\mathbf{v}) = [p_{ij}]_{i,j \in \mathcal{N}_0}$. After $T$ time steps, the estimation error for an estimator $\widehat{\mathbf{p}}$ is defined as

$$e_T(\widehat{\mathbf{p}}, \mathbf{p}(\mathbf{v})) = \mathbb{E}\left[\|\widehat{\mathbf{p}} - \mathbf{p}(\mathbf{v})\|_\infty\right].$$

The following theorem gives the Pareto optimal trade-off between regret and this estimation error.

**Theorem 5.1.** *For any trade-off parameter $\alpha \in [0, 1/2]$, let $\pi$ be the policy from Algorithm 1 run for $T$ steps, and let $\widehat{\mathbf{v}}$ be its output. Denote $\widehat{\mathbf{p}} = \mathbf{p}(\widehat{\mathbf{v}})$ as the plug-in estimator of $\mathbf{p}$. For any instance where $v_i \in [\delta, V]$ for all $i \in [N]$ and some $\delta > 0$, the pair $(\pi, \widehat{\mathbf{p}})$ is Pareto optimal. Moreover,*

*(i) The estimation error is bounded by: $e_T(\widehat{\mathbf{p}}, \mathbf{p}(\mathbf{v})) \leq \dfrac{C_2 V^{5/2}}{2\delta^2} \sqrt{\dfrac{N^{1-\alpha} \log N}{T^{1-\alpha}}}.$*

*(ii) The regret is bounded by: $\mathrm{Reg}_T(\pi, \mathbf{v}) \leq C_3 V N^\alpha T^{1-\alpha} \log^2 NT.$*

*Here, $C_2, C_3 > 0$ are universal constants.*

**Joint Assortment and Pricing Optimization.** We consider the joint assortment and pricing problem, following the formulation of Miao & Chao (2021). At each time step $t$, the seller decides both the assortment $S_t \subseteq \mathcal{S}$ and the price $\boldsymbol{p}_t = [p_{ti}]_{i=1}^N \in \mathbb{R}_+^N$. The customer's choice is governed by a price-sensitive MNL model, where the attraction parameter for item $i$ is a function of its price $p_{ti}$:

$$v_i = \exp(\alpha_i - \beta_i p_{ti}).$$

Here, $\alpha_i > 0$ is the intrinsic quality and $\beta_i > 0$ is the price sensitivity. The seller's objective is to maximize the cumulative expected revenue. For a given assortment $S_t$ and prices $\boldsymbol{p}_t$, the expected revenue is $R(S_t, \boldsymbol{p}_t) := \sum_{j \in S_t} (r_j + p_{tj}) \mathbb{P}(j|S_t, \boldsymbol{p}_t)$. Decisions are constrained to feasible sets, with $S_t \in \mathcal{S}$ and prices $p_{ti} \in [\underline{p}, \bar{p}]$ for a given price range. This formulation generalizes the assortment selection problem; setting all $\beta_i \equiv 0$ reduces it to the previous setting.

To analyze this trade-off, we define the vector of unknown *demand parameters* as $\boldsymbol{\theta} := (\boldsymbol{\alpha}, \boldsymbol{\beta})$, where $\boldsymbol{\alpha} = [\alpha_i]_{i=1}^N$ and $\boldsymbol{\beta} = [\beta_i]_{i=1}^N$. Let $(S^\star, \boldsymbol{p}^\star) := \mathrm{argmax}_{S \in \mathcal{S}, \boldsymbol{p} \in [\underline{p}, \bar{p}]^N} R(S, \boldsymbol{p})$ denote the optimal static policy. The regret and estimation error are then defined as follows:

$$\mathrm{Reg}_T(\pi, \boldsymbol{\theta}) := \sum_{t=1}^T R(S^\star, \boldsymbol{p}^\star) - \mathbb{E}[R(S_t, \boldsymbol{p}_t)], \quad e_T(\widehat{\boldsymbol{\theta}}, \boldsymbol{\theta}) := \mathbb{E}[\|\boldsymbol{\alpha} - \widehat{\boldsymbol{\alpha}}\|_\infty + \|\boldsymbol{\beta} - \widehat{\boldsymbol{\beta}}\|_\infty].$$

The following theorem gives the Pareto optimal trade-off between regret and estimation error.

**Theorem 5.2.** *For any trade-off parameter $\alpha \in [0, 1/2]$, let $(\pi, \widehat{\boldsymbol{\theta}})$ be the policy and estimator pair from Algorithm 5. This pair is Pareto optimal. Moreover, for any environment $\boldsymbol{\theta}$, the following bounds hold:*

*(i) The estimation error is bounded by: $e_T(\widehat{\boldsymbol{\theta}}, \boldsymbol{\theta}) \leq C_5 \sqrt{\dfrac{N^{1-\alpha} \log N}{T^{1-\alpha}}}.$*

*(ii) The regret is bounded by: $\mathrm{Reg}_T(\pi, \boldsymbol{\theta}) \leq C_6 N^\alpha T^{1-\alpha} \log NT.$*

*Here, $C_5, C_6 > 0$ are constants that depend only on the problem parameters.*

## 6 CONCLUSIONS

We characterize the trade-off between regret minimization and parameter estimation in capacitated MNL bandits. First, we establish a tight minimax lower bound of $\Omega(\sqrt{N})$ on the product of estimation error and the square root of regret, resolving the sharp dependence on the number of items $N$. Second, we propose a novel, Pareto-optimal two-phase algorithm that achieves this bound. Our algorithm navigates the Pareto frontier and improves the state-of-the-art upper bound for the uncapacitated case to the optimal $\widetilde{\mathcal{O}}(\sqrt{N})$ rate. Finally, we extend our framework to provide the first Pareto-optimality analyses for customer preference estimation and joint assortment and pricing optimization.

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

CONTENTS

DETAILS OF LLM USAGE

In writing this paper, the LLM was applied to polish our sentences and correct potential typos.

## A  COUNTEREXAMPLE TO THE PRIOR TRADE-OFF LOWER BOUND

This section is dedicated to constructing a counterexample to Simchi-Levi & Wang (2025, Lemma 1). We adopt the notation from the original paper for consistency. We consider a multi-armed bandit problem with $K = 2$ arms, a time horizon of $n$, and an online policy $\pi$. Let $\mathcal{E}_0$ be the set of all bandit instances with rewards in $[-1, 1]$.[3] For any instance $\nu \in \mathcal{E}_0$, let $\mu_i$ be the mean reward of arm $i$. The Average Treatment Effect (ATE) is defined as $\Delta_\nu = \mu_1 - \mu_2$. Let $\widehat{\Delta}_n$ be an estimator for the ATE, and let $\mathcal{R}_\nu(n, \pi)$ be the cumulative regret of policy $\pi$ on instance $\nu$. Lemma 1 in Simchi-Levi & Wang (2025) states the following:

**Lemma A.1.** *When $K = 2$, for any given online decision-making policy $\pi$, the error of any ATE estimators can be lower bounded as follows, for any function $\phi : \mathbb{N} \to [0, 1/4]$ and any $u \in \mathcal{E}_0$.*

$$\inf_{\widehat{\Delta}_n} \max_{\nu \in \mathcal{E}_0} \mathbb{P}_\nu \left( \left| \widehat{\Delta}_n - \Delta_\nu \right| \geq \phi(n) \right) \geq \frac{1}{2} \left[ 1 - \sqrt{\frac{16}{3} \phi(n)^2 \frac{\mathcal{R}_u(n, \pi)}{|\Delta_u|}} \right].$$

To construct a counterexample, we define a specific policy $\pi_m$ for some integer $m$ to be chosen later, an estimator $\widehat{\Delta}_n^\star$, an instance $u^\star$, and a function $\phi^\star(n)$. We then show that for this configuration, the estimation error of our chosen estimator is strictly smaller than the lower bound claimed in Lemma A.1, leading to a contradiction. Specifically, we will show that for all instances $\nu \in \mathcal{E}_0$,

$$\mathbb{P}_\nu \left( \left| \widehat{\Delta}_n^\star - \Delta_\nu \right| \geq \phi^\star(n) \right) < \frac{1}{2} \left[ 1 - \sqrt{\frac{16}{3} \phi^\star(n)^2 \frac{\mathcal{R}_{u^\star}(n, \pi_m)}{|\Delta_{u^\star}|}} \right], \tag{3}$$

which would invalidate the lemma, since the infimum over all estimators must be less than or equal to the value for our specific estimator.

**Intuition of the counterexample.**   The counterexample exploits a flaw in the lemma's premise: the requirement that the lower bound must hold for an arbitrary instance $u \in \mathcal{E}_0$ is overly restrictive. This allows us to construct a policy $\pi_m$ and a specific instance $u^*$ that generates artificially low regret, leading to an inflated and invalid lower bound on the estimation error.

The mechanism can be summarized in two steps:

- **Creating a low-regret scenario:** A specific instance $u^*$ is chosen to compute the regret term $\mathcal{R}_{u^*}(n, \pi_m)$. This instance is tailored to activate the policy's trigger, which causes the algorithm to cease exploration prematurely and thereby incur minimal regret. When this artificially low regret value is substituted into the lemma's inequality, it yields a deceptively high lower bound on the potential estimation error.

- **Achieving low estimation error:** The estimation error, however, is evaluated in a worst-case sense over all instances $\nu \in \mathcal{E}_0$. The policy's trigger is constructed to be a low-probability event for any arbitrary instance. Consequently, for the general class of instances, the policy is forced into a high-exploration mode, guaranteeing that a large number of samples are collected for all arms. This extensive data collection allows a simple empirical mean estimator to achieve a very low estimation error, which is subsequently shown to be smaller than the high lower bound predicted by the lemma.

**Formal construction of the counterexample.**   For simplicity, let $n$ be an even integer. Given an integer $m$ such that $1 \leq m \leq n/2$, we define the policy $\pi_m$ as follows:

---

[3]This is equivalent to our setting with rewards in $[0, 1]$ via a linear transformation.

---

**Phase 1 (First $2m$ steps):** Pull each of the two arms $m$ times.

**Phase 2 (Remaining $n - 2m$ steps):** If every reward arm 2 was $-1$ in Phase 1, pull arm 1 for all remaining steps. Otherwise, pull arm 1 and arm 2 an equal number of times, i.e., $(n-2m)/2$ times each.

---

We select the instance $u^\star \in \mathcal{E}_0$ with deterministic rewards $\mu_1 = 1$ and $\mu_2 = -1$. For this instance, the condition for Phase 2 of policy $\pi_m$ is met, so arm 1 is pulled for the remaining $n - 2m$ steps. The total number of pulls for the suboptimal arm 2 is $m$. The ATE is $\Delta_{u^\star} = \mu_1 - \mu_2 = 2$, and the cumulative regret is $\mathcal{R}_{u^\star}(n, \pi_m) = m \cdot (\mu_1 - \mu_2) = 2m$. Substituting these into the right-hand side of Equation (3) gives

$$\text{RHS} = \frac{1}{2} \left[ 1 - \sqrt{\frac{16}{3} m \phi(n)^2} \right]. \tag{4}$$

Now, for any instance $\nu \in \mathcal{E}_0$, we define an estimator for the ATE based on the empirical means of the rewards from each arm:

$$\widehat{\Delta}_n^\star := \bar{\mu}_1 - \bar{\mu}_2,$$

where $\bar{\mu}_i = \frac{1}{n_i} \sum_{t=1}^{n_i} r_{i,t}$ is the empirical mean reward for arm $i$, and $n_i$ is the total number of times arm $i$ is pulled under instance $\nu$. The error of this estimator is bounded by the triangle inequality: $|\widehat{\Delta}_n^\star - \Delta_\nu| = |(\bar{\mu}_1 - \mu_1) - (\bar{\mu}_2 - \mu_2)| \leq |\bar{\mu}_1 - \mu_1| + |\bar{\mu}_2 - \mu_2|$. By Freedman's inequality, for any instance $\nu \in \mathcal{E}_0$ and for each arm $i \in \{1, 2\}$, with probability at least $1 - \delta/2$, we have

$$|\bar{\mu}_i - \mu_i| \leq \sqrt{\frac{2\sigma_i^2 \log(4/\delta)}{n_i}} + \frac{2 \log(4/\delta)}{3 n_i}, \tag{5}$$

We propose the following claim, which establishes bounds on the estimation error for each arm under policy $\pi_m$.

**Claim A.1.** *Suppose that $m \leq \sqrt{n}$. Under policy $\pi_m$, for any instance $\nu \in \mathcal{E}_0$, conditional on Equation* (5) *holding for both arms, the following error bounds hold:*

*(i) For arm 1, the error is bounded by*

$$|\bar{\mu}_1 - \mu_1| \leq \frac{4 \log(4/\delta)}{m}.$$

*(ii) For arm 2, with probability at least $1 - \delta/4$, the error is bounded by*

$$|\bar{\mu}_2 - \mu_2| \leq \frac{4 \log(4/\delta)}{m}.$$

We will prove this claim in Section A.1. Equation (5) and Claim A.1 together imply that, with probability at least $1 - 5\delta/4$, the estimation error of our proposed estimator is bounded by

$$|\widehat{\Delta}_n^\star - \Delta_\nu| \leq |\bar{\mu}_1 - \mu_1| + |\bar{\mu}_2 - \mu_2| \leq \frac{8 \log(4/\delta)}{m}.$$

Choosing $\phi^\star(n) = \frac{8 \log(4/\delta)}{m}$ for some integer $m \geq 32 \log(4/\delta)$, we have

$$\mathbb{P}_\nu \left( \left| \widehat{\Delta}_n^\star - \Delta_\nu \right| \geq \phi^\star(n) \right) \leq \frac{5\delta}{4}.$$

Equation (4) becomes

$$\text{RHS} = \frac{1}{2} \left[ 1 - \sqrt{\frac{16}{3} m \phi^\star(n)^2} \right] = \frac{1}{2} \left[ 1 - \sqrt{\frac{16}{3} m \cdot \frac{64 \log^2(4/\delta)}{m^2}} \right] = \frac{1}{2} \left[ 1 - \frac{32 \log(4/\delta)}{\sqrt{3m}} \right].$$

To invalidate Lemma A.1, we need to show that

$$\frac{5\delta}{4} < \frac{1}{2}\left[1 - \frac{32\log(4/\delta)}{\sqrt{3m}}\right].$$

When $n$ is sufficiently large, we can choose $m = \lfloor\sqrt{n}\rfloor$ such that $m \geq 3 \cdot (32\log(4/\delta))^2$ for $\delta < 4/15$. This finishes the construction of the counterexample.

**Remark A.1.** *Indeed, our construction can be extended to provide a more powerful refutation. For any positive sequence $\{c_n\}_{n=1}^{\infty}$ that converges to zero, we can show that for a sufficiently large $n$, there exist a policy $\pi_m$, an estimator $\widehat{\Delta}_n^{\star}$, an instance $u^{\star}$, and a function $\phi^{\star}(n)$ such that for all instances $\nu \in \mathcal{E}_0$,*

$$\mathbb{P}_{\nu}\left(\left|\widehat{\Delta}_n^{\star} - \Delta_{\nu}\right| \geq \phi^{\star}(n)\right) < c_n\left[1 - \sqrt{\frac{16}{3}\phi^{\star}(n)^2 \frac{\mathcal{R}_{u^{\star}}(n, \pi_m)}{|\Delta_{u^{\star}}|}}\right].$$

*This demonstrates that the lemma would remain invalid even if the leading constant $1/2$ were replaced by any positive sequence converging to zero.*

### A.1  PROOF OF CLAIM A.1

We prove each part of the claim separately.

**Part (i): Bounding the error for arm 1.**  Since rewards are bounded in $[-1, 1]$, Popoviciu's inequality implies that the variance is bounded by $\sigma_1^2 \leq 1$. By the design of policy $\pi_m$, arm 1 is pulled at least $n/2$ times, so $n_1 \geq n/2$ for any instance $\nu$. Then Equation (5) implies

$$|\bar{\mu}_1 - \mu_1| \leq \sqrt{\frac{2\sigma_1^2\log(4/\delta)}{n_1}} + \frac{2\log(4/\delta)}{3n_1} \leq \sqrt{\frac{4\log(4/\delta)}{n}} + \frac{4\log(4/\delta)}{3n}.$$

Substituting $m \leq \sqrt{n}$ into the inequality gives

$$|\bar{\mu}_1 - \mu_1| \leq \sqrt{\frac{4\log(4/\delta)}{m^2}} + \frac{4\log(4/\delta)}{3m^2} \leq \frac{4\log(4/\delta)}{m},$$

which proves the first part of the claim.

**Part (ii): Bounding the error for arm 2.**  We consider two cases based on the variance $\sigma_2^2$ of the rewards from arm 2.

CASE 1: SMALL VARIANCE.  Suppose that $\sigma_2^2 \leq \frac{4\log(4/\delta)}{m}$. Under policy $\pi_m$, arm 2 is pulled at least $m$ times, so $n_2 \geq m$. Then Equation (5) gives

$$|\bar{\mu}_2 - \mu_2| \leq \sqrt{\frac{2\sigma_2^2\log(4/\delta)}{n_2}} + \frac{2\log(4/\delta)}{3n_2} \leq \sqrt{\frac{8\log^2(4/\delta)}{m^2}} + \frac{2\log(4/\delta)}{3m} \leq \frac{4\log(4/\delta)}{m}.$$

CASE 2: LARGE VARIANCE.  Suppose that $\sigma_2^2 > \frac{4\log(4/\delta)}{m}$. Let $X$ be a random variable for a single reward from arm 2, and let $p := \mathbb{P}(X = -1)$. We can relate $p$ to the variance $\sigma_2^2$ as follows:

$$\sigma_2^2 = \mathbb{E}[X^2] - (\mathbb{E}[X])^2 \leq \mathbb{E}[X^2] + 2\mathbb{E}[X] + 1 = \mathbb{E}[(X+1)^2],$$

where the inequality follows from the fact that $(\mathbb{E}[X] + 1)^2 \geq 0$. We further bound $\mathbb{E}[(X+1)^2]$ in terms of $p$:

$$\mathbb{E}[(X+1)^2] = \mathbb{E}[(X+1)^2|X = -1] \cdot p + \mathbb{E}[(X+1)^2|X > -1] \cdot (1-p).$$

Since $X + 1 = 0$ when $X = -1$ and $(X+1)^2 \leq 4$ when $X \in (-1, 1]$, this simplifies to

$$\sigma_2^2 \leq 4(1-p), \quad \text{which implies} \quad p \leq 1 - \frac{\sigma_2^2}{4}.$$

Since the event $\{n_2 = m\}$ occurs if and only if all $m$ rewards from arm 2 in Phase 1 are $-1$, the probability of $\{n_2 = m\}$ is $p^m$. Consequently, we have

$$\mathbb{P}(n_2 = m) = p^m \leq \left(1 - \frac{\sigma_2^2}{4}\right)^m \leq \exp\left(-\frac{m\sigma_2^2}{4}\right).$$

Using the assumption for this case, $\sigma_2^2 > \frac{4\log(4/\delta)}{m}$, we have

$$\mathbb{P}(n_2 = m) < \exp\left(-\frac{m}{4} \cdot \frac{4\log(4/\delta)}{m}\right) = \exp\left(-\log(4/\delta)\right) = \frac{\delta}{4}.$$

This shows that the "bad event" $\{n_2 = m\}$ occurs with probability less than $\delta/4$. With the complementary probability of at least $1 - \delta/4$, we have $n_2 = n/2$. In this high-probability event, since rewards are bounded in $[-1, 1]$, Popoviciu's inequality implies that the variance is also bounded by $\sigma_2^2 \leq 1$. Using Equation (5) again with $n_2 = n/2$, $\sigma_2^2 \leq 1$, and $m \leq \sqrt{n}$, we have

$$|\bar{\mu}_2 - \mu_2| \leq \sqrt{\frac{2\sigma_2^2\log(4/\delta)}{n_2}} + \frac{2\log(4/\delta)}{3n_2} \leq \sqrt{\frac{4\log(4/\delta)}{n}} + \frac{4\log(4/\delta)}{3n} \leq \sqrt{\frac{4\log(4/\delta)}{m^2}} + \frac{4\log(4/\delta)}{3m^2} \leq \frac{4\log(4/\delta)}{m}.$$

Combining both cases, we conclude that with a total probability of at least $1 - \delta/4$, the error for arm 2 is bounded as claimed. This completes the proof. $\qquad\square$

# B PROOFS FOR SECTION 3: FUNDAMENTAL LIMITS

## B.1 PROOF OF THEOREM 3.1

The proof relies on the following two technical lemmas, which are standard results from the literature. For completeness, their proofs are provided in Section B.4.1 and Section B.4.2.

**Lemma B.1** (Lemma 15.1, Lattimore & Szepesvári (2020)). *For a given assortment $S_t$, define the distribution of customer choices under attraction vector $\mathbf{v}$ and $\mathbf{v}'$ as $\mathbb{P}_{\mathbf{v}}\left(\cdot \mid S_t\right)$ and $\mathbb{P}_{\mathbf{v}'}\left(\cdot \mid S_t\right)$, respectively. Then*

$$D_{KL}\left(\mathbb{P}_{\mathbf{v}}\|\mathbb{P}_{\mathbf{v}'}\right) = \sum_{t=1}^{T} \mathbb{E}_{\mathbf{v}}\left[D_{KL}\left(\mathbb{P}_{\mathbf{v}}\left(\cdot \mid S_t\right)\|\mathbb{P}_{\mathbf{v}'}\left(\cdot \mid S_t\right)\right)\right],$$

*where the expectation is taken over the randomness in the assortments $S_t$ chosen by policy $\pi$.*

**Lemma B.2** (Lemma 3, Chen & Wang (2018)). *Fix an assortment $S_t$ and let $p_i = \mathbb{P}_{\mathbf{v}}\left(c_t = i \mid S_t\right)$ and $q_i = \mathbb{P}_{\mathbf{v}'}\left(c_t = i \mid S_t\right)$ for $i = 0, \ldots, N$. Then*

$$D_{KL}\left(\mathbb{P}_{\mathbf{v}}\left(\cdot \mid S_t\right)\|\mathbb{P}_{\mathbf{v}'}\left(\cdot \mid S_t\right)\right) \leq \sum_{i \in S_t \cup \{0\}} \frac{(p_i - q_i)^2}{q_i}.$$

With these lemmas in place, we are now ready to prove Theorem 3.1.

*Proof.* Fix an admissible pair $(\pi, \widehat{\mathbf{v}})$ and choose an arbitrary revenue vector $\mathbf{r} = (r_1, \ldots, r_N) \in [0, 1]^N$. We start with the first attraction vector instance defined as

$$\mathbf{v} = (v_1, \ldots, v_N) = (1, \ldots, 1).$$

This vector and policy $\pi$ give rise to the distribution $\mathbb{P}_{\mathbf{v}}$, and expectation under $\mathbb{P}_{\mathbf{v}}$ is denoted as $\mathbb{E}_{\mathbf{v}}$. To construct the second attraction vector instance, we first define

$$W_i = \sum_{t=1}^{T} \frac{\mathbb{1}\{i \in S_t\}}{1 + |S_t|}, \tag{6}$$

which denotes the weighted number of times product $i$ is included in the assortments $S_t$ over the time horizon $T$, for each $1 \leq i \leq N$. Let $j$ be the product that is offered least frequently in expectation:

$$j = \underset{1 \leq i \leq N}{\operatorname{argmin}} \mathbb{E}_{\mathbf{v}}[W_i]. \tag{7}$$

The second attraction vector instance is defined as

$$\mathbf{v}' = \begin{cases} 1 + \varepsilon, & \text{if } i = j, \\ 1, & \text{otherwise.} \end{cases}$$

This induces the distribution $\mathbb{P}_{\mathbf{v}'}$. Since $\|\mathbf{v} - \mathbf{v}'\|_\infty = \varepsilon$, Le Cam's method gives

$$\inf_{(\pi, \widehat{\mathbf{v}})} \sup_{(\mathbf{v}, \mathbf{r}) \in \mathcal{E}} e_T(\widehat{\mathbf{v}}, \mathbf{v}) \geq \frac{\varepsilon}{4} \left[ 1 - \|\mathbb{P}_{\mathbf{v}} - \mathbb{P}_{\mathbf{v}'}\|_{\mathrm{TV}} \right].$$

By Pinsker's inequality, we have

$$\|\mathbb{P}_{\mathbf{v}} - \mathbb{P}_{\mathbf{v}'}\|_{\mathrm{TV}} \leq \sqrt{\frac{1}{2} \mathrm{D}_{\mathrm{KL}} \left( \mathbb{P}_{\mathbf{v}} \| \mathbb{P}_{\mathbf{v}'} \right)}.$$

Plugging back into the previous display, we obtain

$$\inf_{(\pi, \widehat{\mathbf{v}})} \sup_{(\mathbf{v}, \mathbf{r}) \in \mathcal{E}} e_T(\widehat{\mathbf{v}}, \mathbf{v}) \geq \frac{\varepsilon}{4} \left[ 1 - \sqrt{\frac{1}{2} \mathrm{D}_{\mathrm{KL}} \left( \mathbb{P}_{\mathbf{v}} \| \mathbb{P}_{\mathbf{v}'} \right)} \right]. \tag{8}$$

Now we bound the KL divergence $\mathrm{D}_{\mathrm{KL}} \left( \mathbb{P}_{\mathbf{v}} \| \mathbb{P}_{\mathbf{v}'} \right)$. By Lemma B.1, this can be achieved by first bounding the conditional KL divergence $\mathrm{D}_{\mathrm{KL}} \left( \mathbb{P}_{\mathbf{v}} \left( \cdot \mid S_t \right) \| \mathbb{P}_{\mathbf{v}'} \left( \cdot \mid S_t \right) \right)$ for a fixed assortment $S_t$, and then taking the expectation over the randomness in $S_t$.

To this end, we analyze the conditional KL divergence by considering two cases for a fixed assortment $S_t$: whether it contains the perturbed product $j$ or not. The following claim, which is central to our proof, provides a sharp upper bound on the conditional KL divergence.

**Claim B.1.** *For a fixed assortment $S_t$, the conditional KL divergence can be upper bounded as follows:*

$$D_{KL} \left( \mathbb{P}_{\mathbf{v}} \left( \cdot \mid S_t \right) \| \mathbb{P}_{\mathbf{v}'} \left( \cdot \mid S_t \right) \right) \leq \frac{2\varepsilon^2}{1 + |S_t|} \mathbb{1}\{j \in S_t\}.$$

The proof of this claim, which involves a careful analysis of the change in choice probabilities for each product in the assortment, is deferred to Section B.1.1. This bound is crucial as it shows that the divergence is proportional to $\varepsilon^2$ and inversely related to the size of the assortment offered.

Having established the KL divergence for a fixed assortment $S_t$ in both cases, we now take the expectation over the randomness in $S_t$:

$$\mathbb{E}_{\mathbf{v}} \left[ \mathrm{D}_{\mathrm{KL}} \left( \mathbb{P}_{\mathbf{v}} \left( \cdot \mid S_t \right) \| \mathbb{P}_{\mathbf{v}'} \left( \cdot \mid S_t \right) \right) \right] \leq 2\varepsilon^2 \cdot \mathbb{E}_{\mathbf{v}} \left[ \frac{\mathbb{1}\{j \in S_t\}}{1 + |S_t|} \right].$$

Then Lemma B.1 gives

$$\mathrm{D}_{\mathrm{KL}} \left( \mathbb{P}_{\mathbf{v}} \| \mathbb{P}_{\mathbf{v}'} \right) = \sum_{t=1}^{T} \mathbb{E}_{\mathbf{v}} \left[ \mathrm{D}_{\mathrm{KL}} \left( \mathbb{P}_{\mathbf{v}} \left( \cdot \mid S_t \right) \| \mathbb{P}_{\mathbf{v}'} \left( \cdot \mid S_t \right) \right) \right] \leq 2\varepsilon^2 \cdot \mathbb{E}_{\mathbf{v}} \left[ \sum_{t=1}^{T} \frac{\mathbb{1}\{j \in S_t\}}{1 + |S_t|} \right] = 2\varepsilon^2 \mathbb{E}_{\mathbf{v}}[W_j].$$

By the definition of index $j$ in (7) and pigeonhole principle, we have

$$\mathbb{E}_{\mathbf{v}}[W_j] \leq \frac{1}{N} \sum_{i=1}^{N} \mathbb{E}_{\mathbf{v}}[W_i].$$

Notice that

$$\sum_{i=1}^{N} \mathbb{E}_{\mathbf{v}}[W_i] = \mathbb{E}_{\mathbf{v}} \left[ \sum_{i=1}^{N} \sum_{t=1}^{T} \frac{\mathbb{1}\{i \in S_t\}}{1 + |S_t|} \right]$$

$$= \mathbb{E}_{\mathbf{v}} \left[ \sum_{t=1}^{T} \frac{1}{1 + |S_t|} \sum_{i=1}^{N} \mathbb{1}\{i \in S_t\} \right]$$

$$= \mathbb{E}_{\mathbf{v}} \left[ \sum_{t=1}^{T} \frac{|S_t|}{1 + |S_t|} \right]$$

$$\leq T.$$

Combining the above two displays, we have

$$\mathrm{D}_{\mathrm{KL}}\left(\mathbb{P}_{\mathbf{v}}\|\mathbb{P}_{\mathbf{v}'}\right) = 2\varepsilon^2 \, \mathbb{E}_{\mathbf{v}}[W_j] \le 2\varepsilon^2 \cdot \frac{1}{N}\sum_{i=1}^{N}\mathbb{E}_{\mathbf{v}}[W_i] \le \frac{2T\varepsilon^2}{N}.$$

Plugging this back into Equation (8), we have

$$\inf_{(\pi,\widehat{\mathbf{v}})} \sup_{(\mathbf{v},\mathbf{r})\in\mathcal{E}} e_T(\widehat{\mathbf{v}},\mathbf{v}) \ge \varepsilon\left[1 - \sqrt{\frac{T\varepsilon^2}{N}}\right].$$

We choose

$$\varepsilon = \sqrt{\frac{N}{4T}}.$$

Then

$$\inf_{(\pi,\widehat{\mathbf{v}})} \sup_{(\mathbf{v},\mathbf{r})\in\mathcal{E}} e_T(\widehat{\mathbf{v}},\mathbf{v}) \ge \frac{1}{4}\sqrt{\frac{N}{4T}}\left(1 - \sqrt{1/4}\right) = \frac{1}{16}\sqrt{\frac{N}{T}}.$$

This completes the proof of Theorem 3.1. $\qquad\square$

### B.1.1   PROOF OF CLAIM B.1

For a given assortment $S_t$, let $p_i = \mathbb{P}_{\mathbf{v}}\left(c_t = i \mid S_t\right)$ and $q_i = \mathbb{P}_{\mathbf{v}'}\left(c_t = i \mid S_t\right)$ denote the choice probabilities under attraction vectors $\mathbf{v}$ and $\mathbf{v}'$, respectively. By the definition of the MNL choice model, these probabilities are given by

$$p_i = \begin{cases} \dfrac{v_i}{1 + \sum_{k\in S_t} v_k}, & \text{if } i \in S_t \cup \{0\}, \\ 0, & \text{otherwise,} \end{cases} \quad \text{and} \quad q_i = \begin{cases} \dfrac{v_i'}{1 + \sum_{k\in S_t} v_k'}, & \text{if } i \in S_t \cup \{0\}, \\ 0, & \text{otherwise.} \end{cases}$$

By Lemma B.2, it suffices to bound the $\chi^2$ divergence $\sum_{i\in S_t\cup\{0\}}(p_i - q_i)^2/q_i$. We proceed by considering the following two cases:

**Case 1**   If product $j$ is not offered in $S_t$, then $v_i = v_i' = 1$ for all $i \in S_t \cup \{0\}$, and therefore $p_i = q_i$ for all $i \in S_t \cup \{0\}$. This implies $\mathrm{D}_{\mathrm{KL}}\left(\mathbb{P}_{\mathbf{v}}\left(\cdot \mid S_t\right)\|\mathbb{P}_{\mathbf{v}'}\left(\cdot \mid S_t\right)\right) = 0$.

**Case 2**   If product $j$ is offered in $S_t$, we consider the following two subcases:

**Subcase 2.1: $i = j$.**   In this case, we have

$$p_j = \frac{1}{1 + |S_t|} = \frac{1}{1 + |S_t|},$$
$$q_j = \frac{1 + \varepsilon}{1 + |S_t| + \varepsilon} = \frac{1 + \varepsilon}{1 + |S_t| + \varepsilon}.$$

Therefore it holds that

$$\frac{(p_j - q_j)^2}{q_j} = \frac{\left(\dfrac{1}{1 + |S_t|} - \dfrac{1 + \varepsilon}{1 + |S_t| + \varepsilon}\right)^2}{\dfrac{1 + \varepsilon}{1 + |S_t| + \varepsilon}}$$

$$= \frac{1 + |S_t| + \varepsilon}{1 + \varepsilon}\left(\frac{\varepsilon|S_t|}{(1 + |S_t|)(1 + |S_t| + \varepsilon)}\right)^2$$

$$= \frac{\varepsilon^2|S_t|^2}{(1 + |S_t|)^2(1 + |S_t| + \varepsilon)(1 + \varepsilon)}$$

$$\le \frac{\varepsilon^2}{1 + |S_t|}.$$

**Subcase 2.2: $i \in S_t \cup \{0\} \setminus \{j\}$.** In this case, we have

$$p_i = \frac{1}{1 + |S_t|} = \frac{1}{1 + |S_t|},$$

$$q_i = \frac{1}{1 + |S_t| + \varepsilon} = \frac{1}{1 + |S_t| + \varepsilon}.$$

Therefore it holds that

$$\frac{(p_i - q_i)^2}{q_i} = \frac{\left( \dfrac{1}{1 + |S_t|} - \dfrac{1}{1 + |S_t| + \varepsilon} \right)^2}{\dfrac{1}{1 + |S_t| + \varepsilon}}$$

$$= (1 + |S_t| + \varepsilon) \left( \frac{\varepsilon}{(1 + |S_t|)(1 + |S_t| + \varepsilon)} \right)^2$$

$$= \frac{\varepsilon^2}{(1 + |S_t|)^2 (1 + |S_t| + \varepsilon)}$$

$$\leq \frac{\varepsilon^2}{(1 + |S_t|)^3}.$$

Combining Lemma B.2 with the results from the three subcases, for all assortments $S_t \in \mathcal{S}$ with $j \in S_t$, we have

$$D_{\mathrm{KL}} \left( \mathbb{P}_{\mathbf{v}} \left( \cdot \mid S_t \right) \| \mathbb{P}_{\mathbf{v}'} \left( \cdot \mid S_t \right) \right) \leq \sum_{i \in S_t \cup \{0\}} \frac{(p_i - q_i)^2}{q_i} \leq \underbrace{\frac{\varepsilon^2}{1 + |S_t|}}_{i=j} + \underbrace{|S_t| \cdot \frac{\varepsilon^2}{(1 + |S_t|)^3}}_{i \in S_t \cup \{0\} \setminus \{j\}} \leq \frac{2\varepsilon^2}{1 + |S_t|}.$$

This completes the proof of Claim B.1. $\qquad\square$

**Remark B.1.** *If we change the parameter instance into $\mathbf{v} = (v_1, \ldots, v_N) = (V, \ldots, V)$ for some constant $V > 1$, and define*

$$\mathbf{v}' = \begin{cases} V + \varepsilon, & \text{if } i = j, \\ V, & \text{otherwise.} \end{cases}$$

*accordingly, the same proof technique still applies. In this case, we can show that*

$$D_{KL} \left( \mathbb{P}_{\mathbf{v}} \left( \cdot \mid S_t \right) \| \mathbb{P}_{\mathbf{v}'} \left( \cdot \mid S_t \right) \right) \leq \frac{2\varepsilon^2}{V(1 + V|S_t|)} \mathbb{1}\{j \in S_t\}.$$

*As a result, Theorem 3.1 can be generalized to*

$$\inf_{(\pi, \widehat{\mathbf{v}})} \sup_{(\mathbf{v}, \mathbf{r}) \in \mathcal{E}} e_T(\widehat{\mathbf{v}}, \mathbf{v}) \geq \frac{V}{16} \sqrt{\frac{N}{T}}.$$

*This yields a lower bound with explicit dependence on the scale $V$ of the attraction parameters.*

## B.2    PROOF OF THEOREM 3.2

For notational convenience, we consider $2N$ items. For the first $N$ items, we set their revenues high to control regret, while for the last $N$ items we set their revenues low to force exploration for estimation. Formally, define the revenue vector instance as $\mathbf{r} = (\mathbf{r}_R, \mathbf{r}_E)$, where $\mathbf{r}_R = (1, \ldots, 1) \in [0, 1]^N$ and $\mathbf{r}_E = (0, \ldots, 0) \in [0, 1]^N$.

We use different constructions of attraction-vector instances to separately control regret and estimation error. For the first $N$ items, we adopt the attraction parameter construction from Chen & Wang (2018). Let $\mathcal{S}_K = \{S \subseteq [N] : |S| = K\}$ be the collection of all size-$K$ subsets. For each $S \in \mathcal{S}_K$, define $\mathbf{v}_R^{(S)} = (v_i^{(S)})_{i=1}^N$ by

$$v_i^{(S)} = \begin{cases} \frac{1 + 0.05\sqrt{N/T}}{K}, & i \in S, \\ \frac{1}{K}, & i \notin S. \end{cases}$$

Let $\mathcal{V}_R := \{\mathbf{v}_R^{(S)} : S \in \mathcal{S}_K\}$ be the collection of attraction-vector instances for the first $N$ items. For the last $N$ items, we define two attraction vector instances $\mathbf{v}_E^{(1)}, \mathbf{v}_E^{(2)} \in \mathbb{R}_+^N$ that correspond to $\mathbf{v}$ and $\mathbf{v}'$ in Theorem 3.1, respectively.

Consider any policy $\pi$ that outputs a sequence of assortments $\{S_t\}_{t=1}^T$. We partition the time horizon into two disjoint subsets:

- (i) $\mathcal{T}_R = \{t \in [T] : S_t \subseteq [N]\}$, the rounds in which only products from the first $N$ items (i.e., those with revenue 1) are offered;
- (ii) $\mathcal{T}_E = [T] \setminus \mathcal{T}_R$, the rounds in which at least one of the last $N$ items (i.e., those with revenue 0) is offered.

Let $T_R = |\mathcal{T}_R|$ and $T_E = |\mathcal{T}_E|$ denote the number of rounds in each subset, respectively.

**Case 1:** $T_E = 0$. In this case, the estimation error is a constant $C_E$, since the attraction parameters of the last $N$ products are not estimated at all. The regret lower bound follows directly from Chen & Wang (2018, Theorem 1):

$$\max_{\mathbf{v}_R \in \mathcal{V}_R} \text{Reg}_T(\pi, \mathbf{v}_R) \geq C_R \sqrt{NT},$$

for some constant $C_R > 0$. Moreover, by Assumption 2.1, we have $T > N$. Then

$$\inf_{(\pi, \widehat{\mathbf{v}})} \sup_{(\mathbf{v}, \mathbf{r}) \in \mathcal{E}} e_T(\widehat{\mathbf{v}}, \mathbf{v}) \sqrt{\text{Reg}_T(\pi, \mathbf{v})} \geq C_E \sqrt{C_R \sqrt{NT}} \geq C_E \sqrt{C_R} \cdot \sqrt{N}.$$

**Case 2:** $T_E > 0$. In this case, we focus on the estimation error for items $\{N+1, \ldots, 2N\}$, which is no greater than the overall estimation error. Since these items are only offered in $\mathcal{T}_E$, by Theorem 3.1 and the construction of $\mathbf{v}^{(1)}$ and $\mathbf{v}^{(2)}$, we have

$$\max_{i=1,2} e_T(\widehat{\mathbf{v}}, \mathbf{v}^{(i)}) \geq \max_{i=1,2} e_T(\widehat{\mathbf{v}}_E, \mathbf{v}_E^{(i)}) \geq \frac{1}{16} \sqrt{\frac{N}{T_E}}.$$

For the regret lower bound, note that each step in $\mathcal{T}_E$ incurs linear regret, since at least one offered product has zero revenue, which is significantly suboptimal. Formally, the optimal revenue lower bounded by

$$R(S^*, \mathbf{v}) = \max_{|S| \leq K} \frac{\sum_{j \in S} r_j v_j}{1 + \sum_{j \in S} v_j} \geq \frac{K \cdot 1/K}{1 + K \cdot 1/K} = \frac{1}{2},$$

for all $\mathbf{v} = (\mathbf{v}_R, \mathbf{v}_E^{(i)})$, where $\mathbf{v}_R \in \mathcal{V}_R$ and $i = 1, 2$. For any assortment $S_{\text{subopt}}$ that includes at least one product from $\{N+1, \ldots, 2N\}$, its expected revenue is upper bounded by

$$R(S_{\text{subopt}}, \mathbf{v}) = \frac{\sum_{j \in S_{\text{subopt}}} r_j v_j}{1 + \sum_{j \in S_{\text{subopt}}} v_j} \leq \frac{(K-1) \cdot 1.05/K + 0}{1 + (K-1) \cdot 1.05/K + 1} \leq \frac{1.05}{3.05},$$

for all $\mathbf{v} = (\mathbf{v}_R, \mathbf{v}_E^{(i)})$, where $\mathbf{v}_R \in \mathcal{V}_R$ and $i = 1, 2$. Therefore, the smallest suboptimality gap $\Delta$ is lower bounded by

$$\Delta = R(S^*, \mathbf{v}) - R(S_{\text{subopt}}, \mathbf{v}) \geq \frac{1}{2} - \frac{1.05}{3.05} = \frac{19}{122}.$$

In other words, the total regret incurred in $\mathcal{T}_E$ is lower bounded by $19T_E/122$. Combined with the regret incurred in $\mathcal{T}_R$ from Chen & Wang (2018, Theorem 1), we have

$$\sup_{(\mathbf{v}, \mathbf{r}) \in \mathcal{E}} \text{Reg}_T(\pi, \mathbf{v}) \geq \max_{i=1,2} \text{Reg}_{T_E}(\pi, \mathbf{v}_E^{(i)}) + \max_{\mathbf{v}_R \in \mathcal{V}_R} \text{Reg}_{T_R}(\pi, \mathbf{v}_R) \geq 19T_E/122 + C_R \sqrt{NT_R},$$

Combining these two inequalities, we obtain

$$\inf_{(\pi, \widehat{\mathbf{v}})} \sup_{(\mathbf{v}, \mathbf{r}) \in \mathcal{E}} e_T(\widehat{\mathbf{v}}, \mathbf{v}) \sqrt{\text{Reg}_T(\pi, \mathbf{v})} \geq \frac{1}{16} \sqrt{\frac{N}{T_E}} \sqrt{C_R \sqrt{NT_R} + 19T_E/122} \geq \frac{1}{16} \sqrt{\frac{19}{122}} \cdot \sqrt{N}.$$

Finally, choosing $C = \min\{C_E \sqrt{C_R}, \sqrt{19/122}/16\}$ concludes the proof.

### B.3 Proof of Theorem 3.3

The proof is by contradiction. Suppose an admissible pair $(\pi, \widehat{\mathbf{v}})$ satisfies the upper bound condition

$$\sup_{(\mathbf{v}, \mathbf{r}) \in \mathcal{E}} e_T(\widehat{\mathbf{v}}, \mathbf{v}) \sqrt{\mathrm{Reg}_T(\pi, \mathbf{v})} = \widetilde{\mathcal{O}}(\sqrt{N}),$$

but is not Pareto optimal. By definition, this means there exists another admissible pair $(\pi', \widehat{\mathbf{v}}')$ that Pareto dominates $(\pi, \widehat{\mathbf{v}})$. This implies that

$$\sup_{(\mathbf{v}, \mathbf{r}) \in \mathcal{E}} e_T(\widehat{\mathbf{v}}', \mathbf{v}) \lesssim \sup_{(\mathbf{v}, \mathbf{r}) \in \mathcal{E}} e_T(\widehat{\mathbf{v}}, \mathbf{v}) \quad \text{and} \quad \sup_{(\mathbf{v}, \mathbf{r}) \in \mathcal{E}} \mathrm{Reg}_T(\pi', \mathbf{v}) \lesssim \sup_{(\mathbf{v}, \mathbf{r}) \in \mathcal{E}} \mathrm{Reg}_T(\pi, \mathbf{v}),$$

with at least one of the inequalities being strict. Consequently, the worst-case performance of the pair $(\pi', \widehat{\mathbf{v}}')$ on the trade-off metric must be strictly better than that of $(\pi, \widehat{\mathbf{v}})$:

$$\sup_{(\mathbf{v}, \mathbf{r}) \in \mathcal{E}} e_T(\widehat{\mathbf{v}}', \mathbf{v}) \sqrt{\mathrm{Reg}_T(\pi', \mathbf{v})} < \sup_{(\mathbf{v}, \mathbf{r}) \in \mathcal{E}} e_T(\widehat{\mathbf{v}}, \mathbf{v}) \sqrt{\mathrm{Reg}_T(\pi, \mathbf{v})}.^4$$

Since $(\pi, \widehat{\mathbf{v}})$ satisfies the $\widetilde{\mathcal{O}}(\sqrt{N})$ upper bound, it follows that

$$\sup_{(\mathbf{v}, \mathbf{r}) \in \mathcal{E}} e_T(\widehat{\mathbf{v}}', \mathbf{v}) \sqrt{\mathrm{Reg}_T(\pi', \mathbf{v})} = o(\sqrt{N}).$$

However, this contradicts the minimax lower bound established in Theorem 3.2, which states that for any admissible pair, the worst-case value of this metric must be at least $\Omega(\sqrt{N})$. Therefore, the initial assumption must be false, and $(\pi, \widehat{\mathbf{v}})$ must be Pareto optimal. $\qquad\square$

### B.4 Proof of Technical Lemmas

#### B.4.1 Proof of Lemma B.1

By the definition of KL divergence, we have

$$D_{\mathrm{KL}}\left(\mathbb{P}_{\mathbf{v}} \| \mathbb{P}_{\mathbf{v}'}\right) = \mathbb{E}_{\mathbf{v}}\left[\log\left(\frac{d\mathbb{P}_{\mathbf{v}}}{d\mathbb{P}_{\mathbf{v}'}}\right)\right].$$

Let $\nu$ denote the underlying measure associated with the canonical MNL model. The Radon-Nikodym derivative of $\mathbb{P}_{\mathbf{v}}$ with respect to $\nu$ is given by

$$\frac{d\mathbb{P}_{\mathbf{v}}}{d\nu}(c_1, S_1, \ldots, c_T, S_T) = \prod_{t=1}^{T} \pi_t(S_t \mid c_1, S_1, \ldots, c_{t-1}, S_{t-1}) \, p_{\mathbf{v}}(c_t \mid S_t),$$

where $\pi_t$ is the policy that chooses assortment $S_t$ at time $t$, and $p_{\mathbf{v}}(\cdot \mid S_t)$ is the Radon-Nikodym derivative of $\mathbb{P}_{\mathbf{v}}(\cdot \mid S_t)$ with respect to $\nu$. Similarly,

$$\frac{d\mathbb{P}_{\mathbf{v}'}}{d\nu}(c_1, S_1, \ldots, c_T, S_T) = \prod_{t=1}^{T} \pi_t(S_t \mid c_1, S_1, \ldots, c_{t-1}, S_{t-1}) \, p_{\mathbf{v}'}(c_t \mid S_t).$$

The chain rule of Radon-Nikodym derivatives gives

$$\frac{d\mathbb{P}_{\mathbf{v}}}{d\mathbb{P}_{\mathbf{v}'}} = \frac{d\mathbb{P}_{\mathbf{v}}}{d\nu} \cdot \frac{d\nu}{d\mathbb{P}_{\mathbf{v}'}} = \frac{d\mathbb{P}_{\mathbf{v}}}{d\nu} \cdot \left(\frac{d\mathbb{P}_{\mathbf{v}'}}{d\nu}\right)^{-1} = \prod_{t=1}^{T} \frac{p_{\mathbf{v}}(\cdot \mid S_t)}{p_{\mathbf{v}'}(\cdot \mid S_t)},$$

where the terms involving the policy $\pi_t$ cancel out. Plugging $p_{\mathbf{v}}(\cdot \mid S_t) = d\mathbb{P}_{\mathbf{v}}(\cdot \mid S_t)/d\nu$ and $p_{\mathbf{v}'}(\cdot \mid S_t) = d\mathbb{P}_{\mathbf{v}'}(\cdot \mid S_t)/d\nu$ into this expression and using the chain rule again, we have

$$\frac{d\mathbb{P}_{\mathbf{v}}}{d\mathbb{P}_{\mathbf{v}'}} = \prod_{t=1}^{T} \frac{\mathbb{P}_{\mathbf{v}}(\cdot \mid S_t)}{\mathbb{P}_{\mathbf{v}'}(\cdot \mid S_t)}.$$

---

[4] Here, $<$ denotes that the left-hand side is asymptotically strictly smaller than the right-hand side, ignoring polylogarithmic factors.

Substituting this into the definition of KL divergence, we obtain

$$D_{KL}\left(\mathbb{P}_{\mathbf{v}} \| \mathbb{P}_{\mathbf{v}'}\right) = \mathbb{E}_{\mathbf{v}}\left[\log\left(\prod_{t=1}^{T}\frac{\mathbb{P}_{\mathbf{v}}\left(\cdot \mid S_t\right)}{\mathbb{P}_{\mathbf{v}'}\left(\cdot \mid S_t\right)}\right)\right] = \sum_{t=1}^{T}\mathbb{E}_{\mathbf{v}}\left[\log\left(\frac{\mathbb{P}_{\mathbf{v}}\left(\cdot \mid S_t\right)}{\mathbb{P}_{\mathbf{v}'}\left(\cdot \mid S_t\right)}\right)\right].$$

By the definition of KL divergence, for a fixed assortment $S_t$, we have

$$D_{KL}\left(\mathbb{P}_{\mathbf{v}}\left(\cdot \mid S_t\right) \| \mathbb{P}_{\mathbf{v}'}\left(\cdot \mid S_t\right)\right) = \mathbb{E}_{\mathbf{v}}\left[\log\left(\frac{\mathbb{P}_{\mathbf{v}}\left(\cdot \mid S_t\right)}{\mathbb{P}_{\mathbf{v}'}\left(\cdot \mid S_t\right)}\right) \bigg| S_t\right].$$

Therefore, we have

$$D_{KL}\left(\mathbb{P}_{\mathbf{v}} \| \mathbb{P}_{\mathbf{v}'}\right) = \sum_{t=1}^{T}\mathbb{E}_{\mathbf{v}}\left[\log\left(\frac{\mathbb{P}_{\mathbf{v}}\left(\cdot \mid S_t\right)}{\mathbb{P}_{\mathbf{v}'}\left(\cdot \mid S_t\right)}\right)\right]$$

$$= \sum_{t=1}^{T}\mathbb{E}_{\mathbf{v}}\left[\mathbb{E}_{\mathbf{v}}\left[\log\left(\frac{\mathbb{P}_{\mathbf{v}}\left(\cdot \mid S_t\right)}{\mathbb{P}_{\mathbf{v}'}\left(\cdot \mid S_t\right)}\right) \bigg| S_t\right]\right]$$

$$= \sum_{t=1}^{T}\mathbb{E}_{\mathbf{v}}\left[D_{KL}\left(\mathbb{P}_{\mathbf{v}}\left(\cdot \mid S_t\right) \| \mathbb{P}_{\mathbf{v}'}\left(\cdot \mid S_t\right)\right)\right].$$

This completes the proof. $\square$

### B.4.2 PROOF OF LEMMA B.2

By the definition of KL divergence, we have

$$D_{KL}\left(\mathbb{P}_{\mathbf{v}}\left(\cdot \mid S_t\right) \| \mathbb{P}_{\mathbf{v}'}\left(\cdot \mid S_t\right)\right) = \sum_{i\in S_t\cup\{0\}} p_i\log\frac{p_i}{q_i}.$$

The logarithmic inequality $\log t \leq t - 1$ for $t > 0$ gives

$$\sum_{i\in S_t\cup\{0\}} p_i\log\frac{p_i}{q_i} \leq \sum_{i\in S_t\cup\{0\}} p_i\left(\frac{p_i}{q_i} - 1\right) = \sum_{i\in S_t\cup\{0\}}\frac{p_i^2}{q_i} - \sum_{i\in S_t\cup\{0\}} p_i = \sum_{i\in S_t\cup\{0\}}\frac{p_i^2}{q_i} - 1.$$

Notice that

$$\sum_{i\in S_t\cup\{0\}}\frac{(p_i-q_i)^2}{q_i} = \sum_{i\in S_t\cup\{0\}}\frac{p_i^2}{q_i} - \sum_{i\in S_t\cup\{0\}} 2p_i + \sum_{i\in S_t\cup\{0\}} q_i = \sum_{i\in S_t\cup\{0\}}\frac{p_i^2}{q_i} - 1,$$

we arrive at the conclusion that

$$D_{KL}\left(\mathbb{P}_{\mathbf{v}}\left(\cdot \mid S_t\right) \| \mathbb{P}_{\mathbf{v}'}\left(\cdot \mid S_t\right)\right) = \sum_{i\in S_t\cup\{0\}} p_i\log\frac{p_i}{q_i} \leq \sum_{i\in S_t\cup\{0\}}\frac{p_i^2}{q_i} - 1 = \sum_{i\in S_t\cup\{0\}}\frac{(p_i-q_i)^2}{q_i}.$$

This completes the proof. $\square$

## C PROOFS FOR SECTION 4: OPTIMAL ALGORITHM

### C.1 PROOF OF THEOREM 4.1

We begin by establishing some key properties of the estimators. Let $L := \min\left\{m \geq 1 : \sum_{\ell=1}^{m}|\mathcal{E}_\ell| \geq T\right\}$ be the total number of epochs, which is a random variable determined by the time horizon $T$. The following lemma summarizes these properties.

**Lemma C.1.** *The following properties hold:*

  *(i) The estimators $\{\widehat{v}_{i,\ell}\}_{\ell=1}^{L}$ are i.i.d. geometric random variables with parameter $1/(1+v_i)$, for all $i = 1, \ldots, N$.*

  *(ii) $\widehat{v}_{i,\ell}$ is an unbiased estimator of $v_i$, i.e. $\mathbb{E}[\widehat{v}_{i,\ell}] = v_i$, for all $i = 1, \ldots, N$ and $\ell = 1, \ldots, L$.*

(iii) *The estimator $\widehat{v}_{i,\ell}$ is a sub-exponential random variable with parameters $\sigma_i^2 = 2v_i^2$ and $b_i = 2v_i$.*

(iv) *Let $M := \lfloor KL/N \rfloor$ be the number of epochs during which each product is at least offered. Then $\widehat{v}_i$ is a sub-exponential random variable with parameters $\sigma^2 = 2V^2/M$ and $b = 2V/M$ for all $i \in [N]$.*

The proofs for (i) and (ii) can be found in Corollary A.1 of Agrawal et al. (2017), while (iii) and (iv) follow from direct computation. We now present another technical lemma.

**Lemma C.2.** *Recall that $M := \lfloor KL/N \rfloor$ is the number of epochs in which each product is offered. Then we have*

$$M \geq \frac{T}{2NV} \geq \frac{C_1}{2V} \log N.$$

The proof of this lemma is provided in Section C.2. We are now ready to prove Theorem 4.1.

*Proof.* Applying Bernstein's inequality to each individual estimator $\widehat{v}_i$ for $i \in [N]$ gives

$$\mathbb{P}\left(|\widehat{v}_i - v_i| \geq \delta\right) \leq 2\exp\left(-\frac{1}{2}\min\left\{\frac{\delta^2}{\sigma^2}, \frac{\delta}{b}\right\}\right)$$

$$= 2\exp\left(-\frac{M}{2}\min\left\{\frac{\delta^2}{2V^2}, \frac{\delta}{2V}\right\}\right),$$

where the last step uses Lemma C.1. Then we apply a union bound over all $N$ products:

$$\mathbb{P}\left(\|\widehat{\mathbf{v}} - \mathbf{v}\|_\infty \geq \delta\right) \leq 2N\exp\left(-\frac{M}{2}\min\left\{\frac{\delta^2}{2V^2}, \frac{\delta}{2V}\right\}\right). \tag{9}$$

Notice that

$$e_T(\widehat{\mathbf{v}}, \mathbf{v}) = \mathbb{E}\left[\|\widehat{\mathbf{v}} - \mathbf{v}\|_\infty\right] = \int_0^\infty \mathbb{P}\left(\|\widehat{\mathbf{v}} - \mathbf{v}\|_\infty \geq \delta\right) \mathrm{d}\delta.$$

We choose $\delta_0 = 2V\sqrt{\dfrac{\log N}{M}}$ and split the integral into three parts:

$$e_T(\widehat{\mathbf{v}}, \mathbf{v}) = \underbrace{\int_0^{\delta_0} \mathbb{P}\left(\|\widehat{\mathbf{v}} - \mathbf{v}\|_\infty \geq \delta\right) \mathrm{d}\delta}_{(\mathrm{I})} + \underbrace{\int_{\delta_0}^{V} \mathbb{P}\left(\|\widehat{\mathbf{v}} - \mathbf{v}\|_\infty \geq \delta\right) \mathrm{d}\delta}_{(\mathrm{II})} + \underbrace{\int_{V}^{\infty} \mathbb{P}\left(\|\widehat{\mathbf{v}} - \mathbf{v}\|_\infty \geq \delta\right) \mathrm{d}\delta}_{(\mathrm{III})}.$$

Now we bound each part of the integral separately. For part (I), since probability is at most 1, we have

$$(\mathrm{I}) = \int_0^{\delta_0} \mathbb{P}\left(\|\widehat{\mathbf{v}} - \mathbf{v}\|_\infty \geq \delta\right) \mathrm{d}\delta \leq \delta_0 = 2V\sqrt{\frac{\log N}{M}}.$$

For part (II), since $\dfrac{\delta^2}{2V^2} \leq \dfrac{\delta}{2V}$ for $\delta \in [\delta_0, V]$, it follows from Equation (9) that

$$\mathbb{P}\left(\|\widehat{\mathbf{v}} - \mathbf{v}\|_\infty \geq \delta\right) \leq 2N\exp\left(-\frac{M\delta^2}{4V^2}\right), \quad \text{for all } \delta \in [\delta_0, V].$$

Then we can bound part (II) as

$$(\mathrm{II}) \leq 2N\int_{\delta_0}^{V} \exp\left(-\frac{M\delta^2}{4V^2}\right) \mathrm{d}\delta = \frac{4NV}{\sqrt{M}}\int_{\sqrt{\log N}}^{\sqrt{M}/2} e^{-x^2}\mathrm{d}x \leq \frac{4NV}{\sqrt{M}}\int_{\sqrt{\log N}}^{\infty} e^{-x^2}\mathrm{d}x$$

where the equality follows from the change of variable $x = \delta\sqrt{M}/(2V)$. Now we leverage the fact that

$$\int_u^\infty e^{-x^2}\mathrm{d}x \leq \frac{e^{-u^2}}{2u}, \quad \text{for all } u > 0.$$

Therefore

$$(\text{II}) \leq \frac{4NV}{\sqrt{M}} \cdot \frac{e^{-\log N}}{2\sqrt{\log N}} = \frac{2V}{\sqrt{M \log N}}.$$

For part (III), since $\frac{\delta^2}{2V^2} \geq \frac{\delta}{2V}$ for $\delta \geq V$, it follows from Equation (9) that

$$\mathbb{P}\left(\|\widehat{\mathbf{v}} - \mathbf{v}\|_\infty \geq \delta\right) \leq 2N \exp\left(-\frac{M\delta}{4V}\right), \quad \text{for all } \delta \geq V.$$

Then we can bound part (III) as

$$(\text{III}) \leq 2N \int_V^\infty \exp\left(-\frac{M\delta}{4V}\right) \mathrm{d}\delta = \frac{8NV}{M} \int_{M/4}^\infty e^{-x} \mathrm{d}x = \frac{8NV}{M \exp(M/4)}.$$

By Lemma C.2, we have $\exp(M/4) \geq N \exp(C_1/8V)$. Thus

$$(\text{III}) \leq \frac{8V}{\exp(C_1/8V)M} \leq \frac{8V}{\exp(C_1/8)M},$$

where the last inequality comes from $V \geq 1$. Combining all three parts, we have

$$\begin{aligned}
e_T(\widehat{\mathbf{v}}, \mathbf{v}) &\leq 2V\sqrt{\frac{\log N}{M}} + \frac{2V}{\sqrt{M \log N}} + \frac{8V}{\exp(C_1/8V)M} \\
&\leq (4 + 8/\exp(C_1/8))V\sqrt{\frac{\log N}{M}} \\
&\leq \left(4\sqrt{2} + 8\sqrt{2}/\exp(C_1/8)\right) V^{3/2} \sqrt{\frac{N \log N}{T}},
\end{aligned}$$

where the last inequality also follows from Lemma C.2. Setting $C_2 = 4\sqrt{2} + 8\sqrt{2}/\exp(C_1/8)$ completes the proof of Theorem 4.1. $\qquad\square$

## C.2 PROOF OF LEMMA C.2

By the design of the algorithm, it is obvious that $|\mathcal{E}_\ell|$ is a geometric random variable with mean $1 + \sum_{i \in S_\ell} v_i$. By the definition of $L$, we have

$$T \leq \mathbb{E}\left[\sum_{\ell=1}^L |\mathcal{E}_\ell|\right] \leq L\left(1 + \max_{1 \leq \ell \leq L} \sum_{i \in S_\ell} v_i\right).$$

By the product constraint $|S_\ell| \leq K$ and the attraction upper bound $v_i \leq V$ for all $1 \leq i \leq N$, we have

$$\max_{1 \leq \ell \leq L} \sum_{i \in S_\ell} v_i \leq KV.$$

Combining these two inequalities gives

$$L \geq \frac{1 + KV}{T}.$$

Then

$$M = \left\lfloor \frac{KL}{N} \right\rfloor \geq \frac{K}{N} \cdot \frac{T}{1 + KV} - 1 \geq \frac{T}{2NV}.$$

Finally, we use Assumption 2.1 to obtain

$$M \geq \frac{T}{2NV} \geq \frac{C_1}{2V} \log N.$$

This completes the proof of Lemma C.2. $\qquad\square$

---

**Algorithm 3 Function** `regret-capacitated`$(N, \mathbf{r}, T, K)$

---

1: **Input:** number of products $N$, revenue vector $\mathbf{r} = (r_1, \ldots, r_N) \in [0,1]^N$, time horizon $T$, assortment capacity $K$
2: Define $\mathcal{S} = \{S \subseteq [N] : |S| \le K\}$ `// feasible assortments under capacity constraint`
3: $\mathcal{E}_1 = \varnothing$; $v_{i,0}^{\text{UCB}} = 1$, $T_i(1) = 0$ for all $i = 1, \ldots, N$
4: $t = 1$, $\ell = 1$ `// keeps track of the time steps and total number of epochs`
5: **while** $t < T$ **do**
6:    `/* Choose the "best" assortment according to the UCB estimates */`
7:    Select $S_\ell = \text{argmax}_{S \in \mathcal{S}} \sum_{i \in S} r_i v_{i,\ell-1}^{\text{UCB}} / (1 + \sum_{j \in S} v_{j,\ell-1}^{\text{UCB}})$
8:    `/* Force exploration of under-explored products */`
9:    Compute $\widehat{S} = \{i \mid T_i(\ell) < 48 \log(\sqrt{N}\ell + 1)\}$
10:    **if** $S_\ell \cap \widehat{S} \ne \varnothing$ **then** choose $S_\ell \in \mathcal{S}$ such that $S_\ell \subseteq \widehat{S}$ **end if**
11:    **repeat**
12:      Offer assortment $S_t = S_\ell$, and observe customer choice $c_t \in S_\ell \cup \{0\}$
13:      $\mathcal{E}_\ell \leftarrow \mathcal{E}_\ell \cup t, t \leftarrow t + 1$
14:    **until** $c_t = 0$ `// no-purchase happens`
15:    **for** $i \in S_\ell$ **do**
16:      Compute $\widehat{v}_{i,\ell} = \sum_{t \in \mathcal{E}_\ell} \mathbb{1}\{c_t = i\}$ `// number of selections for product i`
17:      Update $\mathcal{T}_i(\ell) = \{\tau \le \ell \mid i \in S_\tau\}$, $T_i(\ell) = |\mathcal{T}_i(\ell)|$ `// epochs with product i offered`
18:      Update $\bar{v}_{i,\ell} = \frac{1}{T_i(\ell)} \sum_{\tau \in \mathcal{T}_i(\ell)} \widehat{v}_{i,\tau}$ `// sample mean of the estimates`
19:      Update $v_{i,\ell}^{\text{UCB}} = \bar{v}_{i,\ell} + \max\left\{\sqrt{\bar{v}_{i,\ell}}, \bar{v}_{i,\ell}\right\} \sqrt{\frac{48 \log(\sqrt{N}\ell+1)}{T_i(\ell)}} + \frac{48 \log(\sqrt{N}\ell+1)}{T_i(\ell)}$
20:    **end for**
21:    $\ell \leftarrow \ell + 1, \mathcal{E}_\ell = \varnothing$
22: **end while**
23: **Output:** sequence of assortments $S_1, \ldots, S_T \subseteq [N]$

---

### C.3 An Epoch-Based UCB Algorithm for Regret Minimization

The regret minimization algorithm, proposed by Agrawal et al. (2017), is engineered to balance the exploration-exploitation trade-off within the capacitated MNL bandit framework through an epoch-based Upper Confidence Bound (UCB) strategy. The core mechanism of the algorithm involves iteratively selecting assortments that are optimistic in terms of expected revenue, based on UCB estimates of the products' attraction parameters. This approach inherently biases the selection towards assortments with high revenue potential.

To counteract the risk of premature exploitation and ensure that all products are adequately sampled, the algorithm incorporates a forced-exploration mechanism. This component periodically overrides the optimistic selection to offer assortments composed exclusively of under-explored products.

The algorithm operates in epochs, where an epoch is defined as a sequence of interactions that concludes upon observing a no-purchase event. At the end of each epoch, the algorithm updates the attraction parameter estimates and their corresponding confidence bounds using the newly collected data. This epoch-based structure allows the algorithm to efficiently leverage customer feedback to systematically reduce uncertainty while maintaining a focus on revenue maximization, thereby ensuring a controlled growth of cumulative regret. There are efficient polynomial time algorithms to solve the static assortment optimization problem in Line 7 under MNL model with known parameters (see Avadhanula et al. (2016); Davis et al. (2014); Rusmevichientong et al. (2010b)).

### C.4 Proof of Theorem 4.2

**Part (i): Attraction estimation error.** The estimation of the attraction vector $\mathbf{v}$ is performed exclusively during the estimation phase, which has a duration of $T_e$. By directly applying Theorem 4.1

with the time horizon replaced by $T_e$, we obtain the upper bound on the estimation error. This establishes the first part of the theorem.

**Part (ii): Regret.** The total regret is decomposed into two parts: the regret from the estimation phase ($T_e$ steps) and the regret from the regret minimization phase ($T - T_e$ steps). During the estimation phase, the immediate regret at each step is at most 1, since revenues are bounded in $[0, 1]$. Therefore, the regret from the estimation phase is at most $T_e$. For the regret minimization phase, Agrawal et al. (2017, Theorem 4) provides a bound of

$$C\sqrt{VN(T - T_e)\log(N(T - T_e))} + C'N\log^2(N(T - T_e)) + C''NV\log(N(T - T_e)),$$

for some universal constants $C, C'$, and $C''$. Combining these components and substituting $T_e = \lceil N^\alpha T^{1-\alpha} \rceil$, the total regret is bounded by

$$\text{Reg}_T(\pi, \mathbf{v}) \le N^\alpha T^{1-\alpha} + 1 + C\sqrt{VNT\log NT} + C'N\log^2 NT + C''NV\log NT.$$

Under Assumption 2.1, we have $N \le T$, which implies that the term $N^\alpha T^{1-\alpha}$ dominates the other terms. Specifically, we can simplify the bound:

$$\text{Reg}_T(\pi, \mathbf{v}) \le N^\alpha T^{1-\alpha} + 1 + CV^{1/2}N^\alpha T^{1-\alpha}\sqrt{\log NT} + C'N^\alpha T^{1-\alpha}\log^2 NT + C''VN^\alpha T^{1-\alpha}\log NT$$

$$\le (1 + C + C' + C'')VN^\alpha T^{1-\alpha}\log^2 NT.$$

Setting $C_3 := 1 + C + C' + C''$ completes the proof of the second part.

**Part (iii): The trade-off metric.** Finally, we combine the bounds on the estimation error and the regret as follows:

$$e_T(\widehat{\mathbf{v}}, \mathbf{v})\sqrt{\text{Reg}_T(\pi, \mathbf{v})} \le C_2 V^{3/2}\sqrt{\frac{N^{1-\alpha}\log N}{T^{1-\alpha}}} \cdot \sqrt{C_3 V N^\alpha T^{1-\alpha}\log^2 NT}.$$

$$= C_2\sqrt{C_3} \cdot V^2\sqrt{N\log N\log^2 NT}.$$

Setting $C_4 = C_2\sqrt{C_3}$ completes the proof. $\qquad\square$

## C.5 Implication for Revenue Estimation

This section provides the details for Remark 4.1, which establishes an improved bound on the revenue estimation error. For any given assortment $S$, the expected revenue is defined as

$$R(S, \mathbf{v}) = \sum_{i \in S} \frac{r_i v_i}{1 + \sum_{k \in S} v_k}.$$

Let $\mathbf{R}(\mathbf{v}) = \{R(S, \mathbf{v})\}_{S \in \mathcal{S}}$ denote the vector of true expected revenues over all feasible assortments. For a corresponding estimator $\widehat{\mathbf{R}}$, the uniform estimation error after $T$ steps is defined as

$$e_T(\widehat{\mathbf{R}}, \mathbf{R}(\mathbf{v})) = \mathbb{E}\left[\sup_{S \in \mathcal{S}} |\widehat{R}(S) - R(S, \mathbf{v})|\right] = \mathbb{E}[\|\widehat{\mathbf{R}} - \mathbf{R}(\mathbf{v})\|_\infty].$$

The following corollary establishes a sharp upper bound on this error, demonstrating that an accurate estimation of the attraction parameters leads to a precise estimation of the revenues.

**Corollary C.1.** *Let $\widehat{\mathbf{v}}$ be the estimator produced by Algorithm 1 with time horizon $T$, and let $\widehat{\mathbf{R}} = \mathbf{R}(\widehat{\mathbf{v}})$ be the corresponding plug-in estimator for the revenues $\mathbf{R}(\mathbf{v})$. For a $K$-capacitated MNL bandit problem with attraction parameters $v_i \in [0, V]$ for all $i \in [N]$, we have:*

$$e_T(\widehat{\mathbf{R}}, \mathbf{R}(\mathbf{v})) \le C_2 K V^{3/2}\sqrt{\frac{N\log N}{T}}.$$

This result significantly improves upon the $\mathcal{O}(N^2\sqrt{N/T})$ bound from prior work (Zuo & Qin, 2025), tightening it to $\widetilde{\mathcal{O}}(K\sqrt{N/T})$. This demonstrates that an accurate estimation of the underlying attraction parameters naturally leads to a more precise estimation of the observable revenues.

This direct relationship between the estimation error of attraction parameters and revenues allows us to extend our analysis of the trade-off between estimation and regret. By integrating the bound from Corollary C.1 with the performance guarantees of Algorithm 2, we can characterize the trade-off between revenue estimation and regret, as formalized in the following corollary.

**Corollary C.2.** *For a $K$-capacitated MNL Bandit problem with a bounded instance $\mathbf{v}$ and a trade-off parameter $\alpha \in [0, 1/2]$, consider the admissible pair $(\pi, \widehat{\mathbf{v}})$ produced by Algorithm 2 at time $T$. Let $\widehat{\mathbf{R}} := \mathbf{R}(\widehat{\mathbf{v}})$ be the plug-in estimator for the revenue. This configuration satisfies the following bounds:*

(i) *The estimation error is bounded by:* $e_T(\widehat{\mathbf{R}}, \mathbf{R}(\mathbf{v})) \leq C_2 K V^{3/2} \sqrt{\dfrac{N^{1-\alpha} \log N}{T^{1-\alpha}}}.$

(ii) *The regret is bounded by:* $Reg_T(\pi, \mathbf{v}) \leq C_3 V N^{\alpha} T^{1-\alpha} \log^2 NT.$

(iii) *The trade-off metric is bounded by:* $e_T(\widehat{\mathbf{R}}, \mathbf{R}(\mathbf{v})) \sqrt{Reg_T(\pi, \mathbf{v})} \leq C_4 K V^2 \sqrt{N \log N \log^2 NT}.$

*Here, $C_2, C_3, C_4$ are universal constants.*

Consequently, the trade-off metric is bounded by $e_T(\widehat{\mathbf{R}}, \mathbf{R}(\mathbf{v})) \sqrt{Reg_T(\pi, \mathbf{v})} = \widetilde{\mathcal{O}}(K\sqrt{N})$. This result represents a substantial improvement over the prior $\mathcal{O}(N^{5/2})$ bound from Zuo & Qin (2025).

### C.5.1 Proof of Corollary C.1

Recall that
$$R(S, \mathbf{v}) = \sum_{i \in S} \frac{r_i v_i}{1 + \sum_{k \in S} v_k}.$$

Direct computation shows that for all $j \in S$, the partial derivative of $R(S, \mathbf{v})$ with respect to $v_j$ is given by

$$
\begin{aligned}
\frac{\partial}{\partial v_j} R(S, \mathbf{v}) &= \frac{\partial}{\partial v_j} \left( \frac{r_j v_j}{1 + \sum_{k \in S} v_k} \right) + \sum_{j \neq i \in S} \frac{\partial}{\partial v_j} \left( \frac{r_i v_i}{1 + \sum_{k \in S} v_k} \right) \\
&= \frac{r_j(1 + \sum_{k \in S} v_k) - r_j v_j}{\left(1 + \sum_{k \in S} v_k\right)^2} + \sum_{j \neq i \in S} \frac{-r_i v_i}{\left(1 + \sum_{k \in S} v_k\right)^2} \\
&= \frac{r_j}{1 + \sum_{k \in S} v_k} - \frac{\sum_{i \in S} r_i v_i}{\left(1 + \sum_{k \in S} v_k\right)^2} \\
&= \frac{r_j - R(S, \mathbf{v})}{1 + \sum_{k \in S} v_k}.
\end{aligned}
$$

For all $j \notin S$, it is clear that $\partial R(S, \mathbf{v})/\partial v_j = 0$. Since $r_j \in [0, 1]$ for all $j \in [N]$, we have $0 \leq R(S, \mathbf{v}) < 1$. This implies that $|r_j - R(S, \mathbf{v})| \leq 1$. The $L^1$ norm of the gradient $\nabla R(S, \mathbf{v})$ is therefore bounded by

$$\|\nabla R(S, \mathbf{v})\|_1 = \sum_{j \in S} \left| \frac{\partial}{\partial v_j} R(S, \mathbf{v}) \right| \leq \sum_{j \in S} \frac{|r_j - R(S, \mathbf{v})|}{1 + \sum_{k \in S} v_k} \leq K.$$

By the mean value theorem, there exists some $\mathbf{c}$ on the line segment connecting $\mathbf{v}$ and $\widehat{\mathbf{v}}$ such that

$$R(S, \mathbf{v}) - R(S, \widehat{\mathbf{v}}) = \nabla R(S, \mathbf{v})|_{\mathbf{c}} \cdot (\mathbf{v} - \widehat{\mathbf{v}}).$$

Taking the absolute value and applying the Hölder's inequality

$$|R(S, \mathbf{v}) - R(S, \widehat{\mathbf{v}})| \leq \|\nabla R(S, \mathbf{v})|_{\mathbf{c}}\|_1 \cdot \|\mathbf{v} - \widehat{\mathbf{v}}\|_\infty \leq K \|\mathbf{v} - \widehat{\mathbf{v}}\|.$$

Since the inequality holds for any assortment $S \in \mathcal{S}$, we have

$$\|\widehat{\mathbf{R}} - \mathbf{R}(\mathbf{v})\|_\infty \leq K \|\widehat{\mathbf{v}} - \mathbf{v}\|_\infty.$$

Taking expectation on both sides gives

$$e_T(\widehat{\mathbf{R}}, \mathbf{R}(\mathbf{v})) \leq K e_T(\widehat{\mathbf{v}}, \mathbf{v}).$$

Applying Theorem 4.1 completes the proof. $\qquad\square$

### C.5.2 PROOF OF COROLLARY C.2

The proof is similar to that of Theorem 4.2, with the key difference being the application of Corollary C.1 to bound the revenue estimation error instead of Theorem 4.1. The detailed derivation is omitted for brevity. □

## D DETAILS OF SECTION 5: PREFERENCE FEEDBACK

This section is devoted to the proof of Theorem 5.1. We begin with an overview of the analysis, followed by detailed proofs of the supporting results.

### D.1 OVERVIEW OF ANALYSIS

The proof of Theorem 5.1 follows a similar line of reasoning to the proof of Pareto optimality for the trade-off between attraction estimation and regret minimization (Theorem 3.3). The core of the argument is to establish a fundamental lower bound on the trade-off between preference estimation error and cumulative regret, and then to demonstrate that our proposed algorithm achieves this bound, thereby proving its Pareto optimality. The proof is structured in three main steps.

First, we establish a minimax lower bound that quantifies the inherent tension between estimating the preference vector $\mathbf{p}(\mathbf{v})$ and minimizing regret. This is formalized in the following lemma, which shows that for any policy, there exists a hard-to-learn instance where the product of estimation error and the square root of regret is lower-bounded.

**Lemma D.1** (Minimax Lower Bound). *For any exploration-exploitation policy $\pi$ and preference estimator $\widehat{\mathbf{p}}$, there exists a hard instance $(\mathbf{v}, \mathbf{r}) \in \mathcal{E}$ such that*

$$e_T(\widehat{\mathbf{p}}, \mathbf{p}(\mathbf{v}))\sqrt{Reg_T(\pi, \mathbf{v})} \geq \frac{C\sqrt{N}}{(1+V)^2},$$

*for some universal constant $C > 0$.*

Second, we show that any algorithm that achieves this fundamental lower bound (up to logarithmic factors) must be Pareto optimal. This means no other algorithm can improve upon one metric (estimation error or regret) without degrading the other.

**Lemma D.2** (Condition for Pareto Optimality). *Suppose that for any fixed $(\mathbf{v}, \mathbf{r}) \in \mathcal{E}$, there exists a policy $\pi$ and an estimator $\widehat{\mathbf{p}}$ such that*

$$e_T(\widehat{\mathbf{p}}, \mathbf{p}(\mathbf{v}))\sqrt{Reg_T(\pi, \mathbf{v})} = \widetilde{\mathcal{O}}(\sqrt{N}).$$

*Then the pair $(\pi, \widehat{\mathbf{p}})$ is Pareto optimal.*

Finally, we demonstrate that the specific strategy of using Algorithm 2 in conjunction with a plug-in estimator for the preference vector indeed achieves the $\widetilde{\mathcal{O}}(\sqrt{N})$ rate. This is accomplished by deriving an upper bound on the preference estimation error, which is linked to the error in estimating the attraction vector $\mathbf{v}$.

**Lemma D.3** (Upper Bound for Plug-in Estimator). *Let $\widehat{\mathbf{v}}$ be the estimator produced by Algorithm 1 with time horizon $T$, and let $\widehat{\mathbf{p}} = \mathbf{p}(\widehat{\mathbf{v}})$ be the corresponding plug-in estimator for the preference vector $\mathbf{p}(\mathbf{v})$. Consider an instance where $v_i \in [\delta, V]$ for some $\delta > 0$ and for all $i \in [N]$. Then the estimation error of the preference vector is bounded by*

$$e_T(\widehat{\mathbf{p}}, \mathbf{p}(\mathbf{v})) \leq \frac{C_2 V^{5/2}}{2\delta^2}\sqrt{\frac{N \log N}{T}}.$$

By combining this upper bound with the regret bounds from Theorem 4.2, we can confirm that our algorithm satisfies the condition in Lemma D.2, thus completing the proof of Theorem 5.1. The proofs of these lemmas are provided in the subsequent sections.

## D.2 Proof of Lemma D.1

The proof relies on connecting the estimation error of the preference vector $\mathbf{p}(\mathbf{v})$ to the estimation error of the attraction vector $\mathbf{v}$. We first establish this connection through the following claim.

**Claim D.1.** *If for all admissible pair $(\pi, \widehat{\mathbf{v}})$, there exists a hard instance $(\mathbf{v}, \mathbf{r}) \in \mathcal{E}$ such that $e_T(\widehat{\mathbf{v}}, \mathbf{v}) \geq (1 + V)^2 \varepsilon$, then for all admissible pair $(\pi, \widehat{\mathbf{p}})$, there exists a hard instance $(\mathbf{v}, \mathbf{r}) \in \mathcal{E}$ such that $e_T(\widehat{\mathbf{p}}, \mathbf{p}(\mathbf{v})) \geq \varepsilon$. In other words, $\min_{(\pi, \widehat{\mathbf{v}})} \max_{(\mathbf{v}, \mathbf{r}) \in \mathcal{E}} e_T(\widehat{\mathbf{v}}, \mathbf{v}) \geq (1 + V)^2 \varepsilon$ implies $\min_{(\pi, \widehat{\mathbf{p}})} \max_{(\mathbf{v}, \mathbf{r}) \in \mathcal{E}} e_T(\widehat{\mathbf{p}}, \mathbf{p}(\mathbf{v})) \geq \varepsilon$.*

The proof of this claim is provided in Section D.2.1. The claim establishes that a lower bound on the estimation error of the attraction vector $\mathbf{v}$ directly implies a lower bound on the estimation error of the preference vector $\mathbf{p}(\mathbf{v})$, scaled by a factor of $(1 + V)^2$. Consequently, we can leverage the same hard instance construction from the proof of Theorem 3.2 to establish the desired lower bound for the preference estimation error. The details are omitted for brevity. $\square$

### D.2.1 Proof of Claim D.1

We prove the contrapositive of the claim:

> If there exists an admissible pair $(\pi, \widehat{\mathbf{p}})$ such that for all $(\mathbf{v}, \mathbf{r}) \in \mathcal{E}$, $e_T(\widehat{\mathbf{p}}, \mathbf{p}(\mathbf{v})) \leq \varepsilon$, then there exists an admissible pair $(\pi, \widehat{\mathbf{v}})$ such that for all $(\mathbf{v}, \mathbf{r}) \in \mathcal{E}$, $e_T(\widehat{\mathbf{v}}, \mathbf{v}) \leq (1 + V)^2 \varepsilon$. In other words, $\min_{(\pi, \widehat{\mathbf{p}})} \max_{(\mathbf{v}, \mathbf{r}) \in \mathcal{E}} e_T(\widehat{\mathbf{p}}, \mathbf{p}(\mathbf{v})) \leq \varepsilon$ implies $\min_{(\pi, \widehat{\mathbf{v}})} \max_{(\mathbf{v}, \mathbf{r}) \in \mathcal{E}} e_T(\widehat{\mathbf{v}}, \mathbf{v}) \leq (1 + V)^2 \varepsilon$.

Suppose there exists an admissible pair $(\pi, \widehat{\mathbf{p}})$ such that for all instances $(\mathbf{v}, \mathbf{r}) \in \mathcal{E}$, the estimation error of the preference vector is bounded by $e_T(\widehat{\mathbf{p}}, \mathbf{p}(\mathbf{v})) \leq \varepsilon$. We will construct an estimator $\widehat{\mathbf{v}}$ for the attraction vector $\mathbf{v}$ and show that its error is bounded by $(1 + V)^2 \varepsilon$.

Recall that the preference probability of product $i$ over the no-purchase option is $p_{i0} = v_i/(v_i + v_0)$. Since $v_0 = 1$ by convention, we have $v_i = p_{i0}/(1 - p_{i0})$. This relationship motivates the following plug-in estimator for $v_i$:

$$\widehat{v}_i := \frac{\widehat{p}_{i0}}{1 - \widehat{p}_{i0}}.$$

Let $f(x) = x/(1 - x)$. Then $\widehat{v}_i = f(\widehat{p}_{i0})$ and $v_i = f(p_{i0})$. The derivative is $f'(x) = 1/(1 - x)^2$. Since $v_i \leq V$, we have $p_{i0} = v_i/(v_i + 1) \leq V/(1 + V)$. This implies that $f'(p_{i0}) \leq (1 + V)^2$. By the mean value theorem, there exists some $\xi$ on the line segment between $p_{i0}$ and $\widehat{p}_{i0}$ such that

$$|\widehat{v}_i - v_i| = |f(\widehat{p}_{i0}) - f(p_{i0})| = |f'(\xi)| \cdot |\widehat{p}_{i0} - p_{i0}|.$$

Assuming $\widehat{p}_{i0}$ is sufficiently close to $p_{i0}$, then the derivative is bounded by $|f'(\xi)| \leq (1 + V)^2$. Thus,

$$|\widehat{v}_i - v_i| \leq (1 + V)^2 |\widehat{p}_{i0} - p_{i0}| \leq (1 + V)^2 \|\widehat{\mathbf{p}} - \mathbf{p}(\mathbf{v})\|_\infty.$$

This inequality holds for all $i \in [N]$, so $\|\widehat{\mathbf{v}} - \mathbf{v}\|_\infty \leq (1 + V)^2 \|\widehat{\mathbf{p}} - \mathbf{p}(\mathbf{v})\|_\infty$. Taking the expectation over all sources of randomness, we get

$$e_T(\widehat{\mathbf{v}}, \mathbf{v}) = \mathbb{E}[\|\widehat{\mathbf{v}} - \mathbf{v}\|_\infty] \leq (1 + V)^2 \, \mathbb{E}[\|\widehat{\mathbf{p}} - \mathbf{p}(\mathbf{v})\|_\infty] = (1 + V)^2 e_T(\widehat{\mathbf{p}}, \mathbf{p}(\mathbf{v})).$$

By our initial assumption, $e_T(\widehat{\mathbf{p}}, \mathbf{p}(\mathbf{v})) \leq \varepsilon$, which implies $e_T(\widehat{\mathbf{v}}, \mathbf{v}) \leq (1 + V)^2 \varepsilon$. This shows that if a good estimator for preferences exists, a good estimator for attraction scores also exists, proving the contrapositive statement. $\square$

## D.3 Proof of Lemma D.2

The proof follows the same structure as that of Theorem 3.3, and we omit the details for brevity. $\square$

## D.4 Proof of Lemma D.3

Recall that $p_{ij} = v_i/(v_i + v_j)$. Direct computation shows that the gradient of $p_{ij}$ with respect to $\mathbf{v}$ is given by

$$\frac{\partial p_{ij}}{\partial v_i} = \frac{v_j}{(v_i + v_j)^2}, \quad \frac{\partial p_{ij}}{\partial v_j} = -\frac{v_i}{(v_i + v_j)^2}, \quad \frac{\partial p_{ij}}{\partial v_k} = 0, \text{ for all } k \neq i, j.$$

Since $v_i, v_j \in [\delta, V]$, the $L^1$ norm of the gradient is bounded by

$$\|\nabla p_{ij}\|_1 = \frac{|v_i| + |v_j|}{(v_i + v_j)^2} \leq \frac{V}{2\delta^2}.$$

By the mean value theorem, there exists some $\mathbf{c}$ on the line segment connecting $\mathbf{v}$ and $\widehat{\mathbf{v}}$ such that

$$p_{ij} - \widehat{p}_{ij} = \nabla p_{ij}|_{\mathbf{c}} \cdot (\mathbf{v} - \widehat{\mathbf{v}}).$$

Taking the absolute value and applying the Hölder's inequality

$$|p_{ij} - \widehat{p}_{ij}| \leq \|\nabla p_{ij}|_{\mathbf{c}}\|_1 \cdot \|\mathbf{v} - \widehat{\mathbf{v}}\|_\infty \leq \frac{V}{2\delta^2}\|\mathbf{v} - \widehat{\mathbf{v}}\|_\infty.$$

Since the inequality holds for any pair of products $i, j \in [N] \cup \{0\}$, we have

$$\|\mathbf{p}(\mathbf{v}) - \widehat{\mathbf{p}}\|_\infty \leq \frac{V}{2\delta^2}\|\widehat{\mathbf{v}} - \mathbf{v}\|_\infty.$$

Taking expectation on both sides gives

$$e_T(\widehat{\mathbf{p}}, \mathbf{p}(\mathbf{v})) \leq \frac{V}{2\delta^2} e_T(\widehat{\mathbf{v}}, \mathbf{v}).$$

Applying Theorem 4.1 completes the proof. $\qquad\square$

### D.5 PROOF OF THEOREM 5.1

The proof of Theorem 5.1 combines the upper bounds on preference estimation error and regret derived for Algorithm 2. We show that this algorithm achieves the minimax lower bound established in Lemma D.1, which, by Lemma D.2, implies its Pareto optimality.

**Part (i): Preference estimation error** The preference vector $\mathbf{p}(\mathbf{v})$ is estimated using a plug-in estimator $\widehat{\mathbf{p}} = \mathbf{p}(\widehat{\mathbf{v}})$, where the attraction vector $\widehat{\mathbf{v}}$ is computed during the estimation phase of Algorithm 2, which has a duration of $T_e = \lceil N^\alpha T^{1-\alpha} \rceil$. By applying Lemma D.3 with the time horizon replaced by $T_e$, we obtain the following upper bound on the preference estimation error:

$$e_T(\widehat{\mathbf{p}}, \mathbf{p}(\mathbf{v})) \leq \frac{C_2 V^{5/2}}{2\delta^2}\sqrt{\frac{N \log N}{T_e}} \leq \frac{C_2 V^{5/2}}{2\delta^2}\sqrt{\frac{N \log N}{N^\alpha T^{1-\alpha}}} = \frac{C_2 V^{5/2}}{2\delta^2}\sqrt{\frac{N^{1-\alpha} \log N}{T^{1-\alpha}}}.$$

**Part (ii): Regret** The regret analysis for Algorithm 2 is identical to that in the proof of Theorem 4.2. The total regret is dominated by the regret incurred during the estimation phase and the regret from the subsequent regret minimization phase. As shown in Theorem 4.2, the cumulative regret is bounded by:

$$\mathrm{Reg}_T(\pi, \mathbf{v}) \leq C_3 V N^\alpha T^{1-\alpha} \log^2 NT.$$

**Part (iii): The trade-off metric and Pareto optimality** Combining the bounds on the preference estimation error and the cumulative regret, we can evaluate the trade-off metric:

$$
\begin{aligned}
e_T(\widehat{\mathbf{p}}, \mathbf{p}(\mathbf{v}))\sqrt{\mathrm{Reg}_T(\pi, \mathbf{v})} &\leq \left(\frac{C_2 V^{5/2}}{2\delta^2}\sqrt{\frac{N^{1-\alpha} \log N}{T^{1-\alpha}}}\right) \cdot \sqrt{C_3 V N^\alpha T^{1-\alpha} \log^2 NT} \\
&= \frac{C_2\sqrt{C_3}V^3}{2\delta^2}\sqrt{(N^{1-\alpha} \log N) \cdot (N^\alpha \log^2 NT)} \\
&= \frac{C_2\sqrt{C_3}V^3}{2\delta^2}\sqrt{N \log N \log^2 NT}.
\end{aligned}
$$

This shows that the product of the preference estimation error and the square root of regret scales as $\widetilde{\mathcal{O}}(\sqrt{N})$. According to Lemma D.2, any admissible pair $(\pi, \widehat{\mathbf{p}})$ that achieves this rate is Pareto optimal. Therefore, the strategy defined by Algorithm 2 and the plug-in preference estimator is Pareto optimal for the trade-off between preference estimation and regret minimization. $\qquad\square$

# E   DETAILS OF SECTION 5: PRICING SETTING

This appendix provides the detailed algorithms and proofs for the pricing setting from Section 5. We present a two-phase algorithm for joint assortment and price optimization and establish its Pareto optimality for the trade-off between parameter estimation error and cumulative regret.

## E.1   THE ALGORITHMIC FRAMEWORK

We first make a standard assumption on the range of the model parameters.

**Assumption E.1.** *There exists $\underline{\alpha} \leq \bar{\alpha}$ and $\underline{\beta} \leq \bar{\beta}$ such that $\alpha_i \in [\underline{\alpha}, \bar{\alpha}]$ and $\beta_i \in [\underline{\beta}, \bar{\beta}]$ for all $i \in [N]$.*

This is a mild assumption common in the literature because in most practical applications the product-specific quality ($\alpha_i$) and price-sensitivity ($\beta_i$) parameters are expected to lie within a bounded range.

### E.1.1   THE EXPLORATION ALGORITHM

Algorithm 4 presents our procedure for estimating the model parameters $\boldsymbol{\theta} = (\boldsymbol{\alpha}, \boldsymbol{\beta})$ in the joint assortment and price optimization setting. This algorithm is analogous to Algorithm 1 for the standard MNL bandits. The design of Algorithm 4 is crucial for disentangling the price-independent quality parameter $\alpha_i$ from the price-sensitivity parameter $\beta_i$. The algorithm operates in epochs, similar to Algorithm 1. In each epoch $\tau$, a fixed assortment $S_\tau$ is offered.

The key difference is the introduction of two distinct pricing phases within each epoch. First, the assortment $S_\tau$ is repeatedly offered with a uniformly high price $\bar{p}$ for all products until a no-purchase event occurs. This phase allows for the estimation of a low attraction score, $v_i^l = \exp(\alpha_i - \beta_i \bar{p})$, for each product $i \in S_\tau$. Subsequently, the same assortment is offered with a uniformly low price $\underline{p}$ until another no-purchase event, which facilitates the estimation of a high attraction score, $v_i^u = \exp(\alpha_i - \beta_i \underline{p})$.

The stopping condition for each phase (observing a no-purchase) is a standard technique that allows the empirical purchase counts to serve as unbiased estimators for the true attraction scores. By obtaining estimates for these two attraction scores under different prices, we create a system of two log-linear equations for each product, which can then be solved to uniquely determine the estimators $\widehat{\alpha}_i$ and $\widehat{\beta}_i$.

### E.1.2   THE MAIN TRADE-OFF ALGORITHM

Algorithm 5 presents our main algorithm for the joint assortment and price optimization problem. It adopts a two-phased structure to manage the trade-off between parameter estimation and regret minimization. The first phase is dedicated to estimation, running for $T_e$ time steps, a duration determined by the trade-off parameter $\alpha$. In this phase, it utilizes Algorithm 4 to compute estimates $\widehat{\boldsymbol{\alpha}}$ and $\widehat{\boldsymbol{\beta}}$. For the subsequent $T - T_e$ time steps, the algorithm transitions to a regret minimization phase, for which it employs the Thompson Sampling-based Price Selection (TS-PS) algorithm from Miao & Chao (2021).

## E.2   OVERVIEW OF ANALYSIS

Our analysis centers on establishing the Pareto optimality of our proposed framework. We first derive a fundamental lower bound on the trade-off between preference estimation error and cumulative regret. We then demonstrate that Algorithm 5 achieves this lower bound.

The lower bound is a direct consequence of the result for the standard MNL bandit problem (Theorem 3.2). Since the joint assortment and price optimization problem is a strict generalization of the standard MNL setting, any lower bound on the latter must also apply to the former.

The main technical challenge is to prove that Algorithm 5 is rate-optimal. This involves analyzing the statistical properties of the parameter estimators from Algorithm 4 and using these properties to bound the cumulative regret incurred during the subsequent regret minimization phase.

---

**Algorithm 4 Function** `price-estimation(N,r,T,K)`

---

1: **Input:** number of products $N$, revenue vector $\mathbf{r} = (r_1, \ldots, r_N) \in [0,1]^N$, time horizon $T$, assortment capacity $K$.
2: $t = 1; \tau = 1$ // keeps track of the time steps and total number of epochs
3: $\mathcal{E}_1^l = \varnothing, \mathcal{E}_1^u = \varnothing; T_i(1) = 0, i = 1, \ldots, N$
4: **while** $t < T$ **do**
5: $\quad$ /* Choose the assortment "evenly" */
6: $\quad$ Select $S_\tau \subseteq [N]$ as the set of $K$ products with the fewest offered epochs $T_i(\tau)$
7: $\quad$ /* offer price $\bar{p}$ to obtain $\widehat{v}_i^l$ */
8: $\quad$ **repeat**
9: $\quad\quad$ Offer assortment $S_t = S_\tau$ with price vector $\boldsymbol{p}_t \equiv \bar{p}$, and observe customer choice $c_t \in S_\tau \cup \{0\}$
10: $\quad\quad$ $\mathcal{E}_\tau^l \leftarrow \mathcal{E}_\tau^l \cup t, t \leftarrow t + 1$
11: $\quad$ **until** $c_t = 0$ // no-purchase happens
12: $\quad$ /* Repeat the same procedure with price $\underline{p}$ to obtain $\widehat{v}_i^u$ */
13: $\quad$ **repeat**
14: $\quad\quad$ Offer assortment $S_t = S_\tau$ with price vector $\boldsymbol{p}_t \equiv \underline{p}$, and observe customer choice $c_t \in S_\tau \cup \{0\}$
15: $\quad\quad$ $\mathcal{E}_\tau^u \leftarrow \mathcal{E}_\tau^u \cup t, t \leftarrow t + 1$
16: $\quad$ **until** $c_t = 0$ // no-purchase happens
17: $\quad$ **for** $i \in S_\tau$ **do**
18: $\quad\quad$ Compute $\widehat{v}_{i,\tau}^l = \sum_{t \in \mathcal{E}_\tau^l} \mathbb{1}\{c_t = i\}$, $\widehat{v}_{i,\tau}^u = \sum_{t \in \mathcal{E}_\tau^u} \mathbb{1}\{c_t = i\}$ // number of selections
19: $\quad\quad$ Update $\mathcal{T}_i(\tau) = \{\eta \leq \tau \mid i \in S_\eta\}, T_i(\tau) = |\mathcal{T}_i(\tau)|$ // epochs with product $i$ offered
20: $\quad\quad$ Update $\widehat{v}_i^l = \frac{1}{T_i(\tau)} \sum_{\tau \in \mathcal{T}_i(\tau)} \widehat{v}_{i,\tau}^l, \widehat{v}_i^u = \frac{1}{T_i(\tau)} \sum_{\tau \in \mathcal{T}_i(\tau)} \widehat{v}_{i,\tau}^u$ // sample mean
21: $\quad$ **end for**
22: $\quad$ $\tau \leftarrow \tau + 1; \mathcal{E}_\tau^l = \varnothing, \mathcal{E}_\tau^u = \varnothing$
23: **end while**
24: /* Use log-valued linear regression to solve for parameters $\widehat{\boldsymbol{\alpha}}, \widehat{\boldsymbol{\beta}}$ */
25: **for** $i = 1$ to $N$ **do**
26: $\quad$ $(\widehat{\alpha}_i, \widehat{\beta}_i) = \operatorname{argmin}_{a,b} \left[ |\log(\widehat{v}_i^u) - (a - b\underline{p})|^2 + |\log(\widehat{v}_i^l) - (a - b\bar{p})|^2 \right]$
27: **end for**
28: **Output:** estimators $\widehat{\boldsymbol{\alpha}} = (\widehat{\alpha}_1, \ldots, \widehat{\alpha}_N), \widehat{\boldsymbol{\beta}} = (\widehat{\beta}_1, \ldots, \widehat{\beta}_N)$, sequence of assortments $S_1, \ldots, S_T \subseteq [N]$, sequence of price vectors $\boldsymbol{p}_1, \ldots, \boldsymbol{p}_T \in [\underline{p}, \bar{p}]^N$

---

**Algorithm 5** Regret-Estimation Error Trade-off for Joint Assortment and Price Optimization

---

1: **Input:** number of products $N$, revenue vector $\mathbf{r} = (r_1, \ldots, r_N) \in [0,1]^N$, time horizon $T$, assortment capacity $K$, trade-off parameter $\alpha \in [0, 1/2]$
2: Calculate estimation steps $T_e = \lceil N^\alpha T^{1-\alpha} \rceil$ // Assumption 2.1 ensures $T_e \leq T$
3: $\widehat{\boldsymbol{\alpha}}, \widehat{\boldsymbol{\beta}}, S_1, \ldots, S_{T_e}, \boldsymbol{p}_1, \ldots, \boldsymbol{p}_{T_e} =$ `price-estimation`$(N, \mathbf{r}, T_e, K)$ // Algorithm 4 for parameter estimation
4: $S_{T_e+1}, \ldots, S_T, \boldsymbol{p}_{T_e+1}, \ldots, \boldsymbol{p}_T =$ `TS-PS`$(N, \mathbf{r}, T - T_e, K)$ // Algorithm 1 in Miao & Chao (2021) for regret minimization

---

The analysis proceeds in two main steps. First, we bound the estimation error of the parameters $(\boldsymbol{\alpha}, \boldsymbol{\beta})$ produced by Algorithm 4. This is achieved by first establishing a concentration inequality for the intermediate attraction score estimators, $\widehat{\mathbf{v}}^l$ and $\widehat{\mathbf{v}}^u$, by adapting known results from the standard MNL bandit literature (Lemma E.1). We then perform a perturbation analysis to show how the error in these attraction score estimates propagates to the final parameter estimates, $(\widehat{\boldsymbol{\alpha}}, \widehat{\boldsymbol{\beta}})$, which are obtained by solving a system of log-linear equations (Lemma E.2).

Second, we bound the regret incurred during the second phase by applying the analysis of the TS-PS algorithm from Miao & Chao (2021). By combining the estimation error from the first phase with

the regret bound from the second, we show that Algorithm 5 achieves the minimax lower bound for a suitable choice of the trade-off parameter $\alpha$. This establishes the Pareto optimality of our proposed framework.

### E.3 Properties of the Estimators

We now analyze the statistical properties of the estimators produced by Algorithm 4. The core of our analysis relies on first bounding the error of the estimated attraction scores, $\widehat{\mathbf{v}}^l$ and $\widehat{\mathbf{v}}^u$, which serve as intermediate quantities. For a fixed price level (either the high price $\bar{p}$ or the low price $\underline{p}$), the estimation problem for the corresponding attraction scores reduces to the standard MNL bandits. This allows us to leverage existing concentration results. The following lemma provides a uniform error bound on these estimators with respect to their true population-level counterparts, $\mathbf{v}^l$ and $\mathbf{v}^u$.

**Lemma E.1.** *Let the true attraction scores be $v_i^u = \exp(\alpha_i - \beta_i \underline{p})$ and $v_i^l = \exp(\alpha_i - \beta_i \bar{p})$ for each product $i \in [N]$. Let $\mathbf{v}^u = (v_1^u, \ldots, v_N^u)$ and $\mathbf{v}^l = (v_1^l, \ldots, v_N^l)$ be the corresponding vectors. The estimators $\widehat{\mathbf{v}}^u = (\widehat{v}_1^u, \ldots, \widehat{v}_N^u)$ and $\widehat{\mathbf{v}}^l = (\widehat{v}_1^l, \ldots, \widehat{v}_N^l)$ obtained from Algorithm 4 satisfy:*

$$\max \left\{ \|\widehat{\mathbf{v}}^u - \mathbf{v}^u\|_\infty, \|\widehat{\mathbf{v}}^l - \mathbf{v}^l\|_\infty \right\} \le C_2 \exp \left( \frac{3}{2}(\bar{\alpha} - \underline{\beta p}) \right) \sqrt{\frac{N \log N}{T}}.$$

*Here, $C_2 > 0$ is a universal constant.*

With the error bounds for the attraction score estimators established in Lemma E.1, we can proceed to analyze the error of the final parameter estimators, $\widehat{\boldsymbol{\alpha}}$ and $\widehat{\boldsymbol{\beta}}$. These estimators are derived from the estimated attraction scores, $\widehat{\mathbf{v}}^u$ and $\widehat{\mathbf{v}}^l$, by solving a system of log-linear equations for each product $i$. The following lemma provides a key perturbation result that bounds the error of the regression output $(\widehat{\alpha}_i, \widehat{\beta}_i)$ in terms of the error in its inputs $(\widehat{v}_i^u, \widehat{v}_i^l)$.

**Lemma E.2.** *Let $(\alpha_i, \beta_i)$ be the true parameters for product $i$. Suppose we have positive estimates for the attraction scores, $\widehat{v}_i^u$ and $\widehat{v}_i^l$, that are $\varepsilon$-close to their true values:*

$$|\widehat{v}_i^u - \exp(\alpha_i - \beta_i \underline{p})| \le \varepsilon, \quad and \quad |\widehat{v}_i^l - \exp(\alpha_i - \beta_i \bar{p})| \le \varepsilon,$$

*where $0 < \varepsilon < \frac{1}{2} \exp(\underline{\alpha} - \bar{\beta}\bar{p})$. Let the estimated parameters $(\widehat{\alpha}_i, \widehat{\beta}_i)$ be the solution to the log-linear regression problem:*

$$(\widehat{\alpha}_i, \widehat{\beta}_i) = \operatorname*{argmin}_{a,b} \left[ |\log(\widehat{v}_i^u) - (a - b\underline{p})|^2 + |\log(\widehat{v}_i^l) - (a - b\bar{p})|^2 \right].$$

*Then, the estimation error is bounded as follows:*

$$|\widehat{\alpha}_i - \alpha_i| + |\widehat{\beta}_i - \beta_i| \le \left( \frac{2 + \bar{p} + \underline{p}}{\bar{p} - \underline{p}} \right) \exp(-\underline{\alpha} + \bar{\beta}\bar{p}) \, \varepsilon.$$

By combining Lemma E.1 and Lemma E.2, we can bound the overall error of the parameter estimators from Algorithm 4. This result is formalized in the following theorem.

**Theorem E.1.** *Let $\boldsymbol{\theta} = (\boldsymbol{\alpha}, \boldsymbol{\beta})$ be the true parameters and $\widehat{\boldsymbol{\theta}} = (\widehat{\boldsymbol{\alpha}}, \widehat{\boldsymbol{\beta}})$ be the estimators produced by Algorithm 4. Under Assumption E.1, the estimation error is bounded as follows:*

$$e_T(\widehat{\boldsymbol{\theta}}, \boldsymbol{\theta}) \le C_5 \sqrt{\frac{N \log N}{T}},$$

*where $C_5 = C_2 \left( \dfrac{2 + \bar{p} + \underline{p}}{\bar{p} - \underline{p}} \right) \exp \left( \frac{3}{2}(\bar{\alpha} - \underline{\beta p}) - \underline{\alpha} + \bar{\beta}\bar{p} \right)$ is a constant that depends only on the problem parameters.*

The proof of Theorem E.1 follows directly from Lemmas E.1 and E.2 by substituting the bound on the attraction score estimation error into the perturbation $\varepsilon$ for the parameter estimators.

### E.3.1 PROOF OF LEMMA E.1

The proof is a direct application of Theorem 4.1. In the setting of Algorithm 4, for a fixed price vector (either $\boldsymbol{p} \equiv \bar{\boldsymbol{p}}$ or $\boldsymbol{p} \equiv \underline{\boldsymbol{p}}$), the model reduces to a standard MNL bandit problem. The attraction scores are constant, given by $v_i^l = \exp(\alpha_i - \beta_i \bar{p})$ and $v_i^u = \exp(\alpha_i - \beta_i \underline{p})$, respectively.

Under Assumption E.1, the maximum possible attraction score across all products and both price levels is bounded by:

$$\max_{i \in [N]} \{v_i^u, v_i^l\} \leq \exp(\bar{\alpha} - \underline{\beta}\underline{p}).$$

This provides the necessary bound on the attraction scores to apply Theorem 4.1. The theorem's result on the concentration of estimators for attraction scores thus holds for both $\widehat{\mathbf{v}}^u$ and $\widehat{\mathbf{v}}^l$, yielding the desired bound. $\qquad\square$

### E.3.2 PROOF OF LEMMA E.2

The solution to the least-squares problem is given by the closed-form expressions:

$$\widehat{\beta}_i = \frac{\log(\widehat{v}_i^u) - \log(\widehat{v}_i^l)}{\bar{p} - \underline{p}}, \quad \text{and} \quad \widehat{\alpha}_i = \log(\widehat{v}_i^u) + \widehat{\beta}_i \underline{p}.$$

The true parameters $(\alpha_i, \beta_i)$ satisfy the same relationships with the true attraction scores:

$$\beta_i = \frac{\log(v_i^u) - \log(v_i^l)}{\bar{p} - \underline{p}}, \quad \text{and} \quad \alpha_i = \log(v_i^u) + \beta_i \underline{p}.$$

Let us define the log-transformed attraction scores as $\widehat{y}_i^u = \log(\widehat{v}_i^u)$, $\widehat{y}_i^l = \log(\widehat{v}_i^l)$, and similarly $y_i^u = \log(v_i^u)$, $y_i^l = \log(v_i^l)$. By the mean value theorem and the given condition on $\varepsilon$, for some $c \in [\min(\widehat{v}, v), \max(\widehat{v}, v)]$, we have

$$|\widehat{y}_i^u - y_i^u| = \left| \frac{1}{c}(\widehat{v}_i^u - v_i^u) \right| \leq \frac{|\widehat{v}_i^u - v_i^u|}{\min\{\widehat{v}_i^u, v_i^u\}} \leq \frac{\varepsilon}{v_i^u - \varepsilon} \leq \frac{2\varepsilon}{v_i^u}.$$

A similar argument holds for the lower price point, yielding

$$|\widehat{y}_i^l - y_i^l| \leq \frac{2\varepsilon}{v_i^l}.$$

Now, we bound the error in the parameter estimates. For the price-sensitivity parameter $\beta_i$:

$$|\widehat{\beta}_i - \beta_i| = \left| \frac{(\widehat{y}_i^u - y_i^u) - (\widehat{y}_i^l - y_i^l)}{\bar{p} - \underline{p}} \right|$$

$$\leq \frac{|\widehat{y}_i^u - y_i^u| + |\widehat{y}_i^l - y_i^l|}{\bar{p} - \underline{p}} \leq \frac{2\varepsilon/v_i^u + 2\varepsilon/v_i^l}{\bar{p} - \underline{p}}.$$

Since $v_i^l \leq v_i^u$, we have $1/v_i^u \leq 1/v_i^l$, which simplifies the bound to:

$$|\widehat{\beta}_i - \beta_i| \leq \frac{4\varepsilon}{(\bar{p} - \underline{p})v_i^l}.$$

For the quality parameter $\alpha_i$, we have

$$|\widehat{\alpha}_i - \alpha_i| = |(\widehat{y}_i^u - y_i^u) + (\widehat{\beta}_i - \beta_i)\underline{p}|$$

$$\leq |\widehat{y}_i^u - y_i^u| + |\widehat{\beta}_i - \beta_i|\underline{p} \leq \frac{2\varepsilon}{v_i^u} + \frac{4\varepsilon\underline{p}}{(\bar{p} - \underline{p})v_i^l}.$$

Again using $v_i^l \leq v_i^u$, we obtain

$$|\widehat{\alpha}_i - \alpha_i| \leq \frac{2\varepsilon}{v_i^l} + \frac{4\varepsilon\underline{p}}{(\bar{p} - \underline{p})v_i^l} = \frac{2\varepsilon}{v_i^l}\left(1 + \frac{2\underline{p}}{\bar{p} - \underline{p}}\right).$$

Combining the error bounds for $\widehat{\alpha}_i$ and $\widehat{\beta}_i$ and using the fact that $v_i^l = \exp(\alpha_i - \beta_i \bar{p}) \geq \exp(\underline{\alpha} - \bar{\beta}\bar{p})$ from Assumption E.1, we obtain the desired result. $\qquad\square$

### E.4 PROOF OF THEOREM 5.2

The lower bound from Theorem 3.2 applies since the pricing problem generalizes the standard MNL bandits. We now prove that Algorithm 5 achieves this bound, which establishes its Pareto optimality. We analyze the three components of the trade-off metric separately: (i) the parameter estimation error, (ii) the cumulative regret, and (iii) the combined trade-off metric.

**Part (i): Parameter estimation error** By applying Theorem E.1 with the time horizon replaced by $T_e = \lceil N^\alpha T^{1-\alpha} \rceil$, we obtain the following upper bound on the parameter estimation error:

$$e_T(\widehat{\boldsymbol{\theta}}, \boldsymbol{\theta}) \leq C_5 \sqrt{\frac{N \log N}{T_e}} \leq C_5 \sqrt{\frac{N \log N}{N^\alpha T^{1-\alpha}}} = C_5 \sqrt{\frac{N^{1-\alpha} \log N}{T^{1-\alpha}}}.$$

**Part (ii): Regret** The total regret is the sum of the regret from the estimation phase (the first $T_e$ steps) and the regret from the regret minimization phase (the remaining $T - T_e$ steps). In the estimation phase, since revenues are bounded in $[0, 1]$, the immediate regret at each step is at most 1. Thus, the regret from this phase is at most $T_e$. For the regret minimization phase, Miao & Chao (2021, Theorem 1) provides a bound of

$$CN \log(N(T - T_e)) + C' \sqrt{N(T - T_e) \log(N(T - T_e))},$$

where $C$ and $C'$ are constants dependent on the problem parameters.[5] Combining these components and substituting $T_e = \lceil N^\alpha T^{1-\alpha} \rceil$, the total regret is bounded by:

$$\mathrm{Reg}_T(\pi, \mathbf{v}) \leq N^\alpha T^{1-\alpha} + 1 + CN \log NT + C' \sqrt{NT \log NT}.$$

Under Assumption 2.1, we have $N \leq T$, which implies that the term $N^\alpha T^{1-\alpha}$ dominates the other terms in the bound. This allows us to simplify the expression to:

$$\mathrm{Reg}_T(\pi, \mathbf{v}) \leq (1 + C + C')N^\alpha T^{1-\alpha} \log NT.$$

By defining a new constant $C_6 := 1 + C + C'$, we obtain the desired bound for the regret.

**Part (iii): The trade-off metric and Pareto optimality** Combining the bounds on the estimation error and the cumulative regret, we can evaluate the trade-off metric:

$$e_T(\widehat{\boldsymbol{\theta}}, \boldsymbol{\theta}) \sqrt{\mathrm{Reg}_T(\pi, \mathbf{v})} \leq \left( C_5 \sqrt{\frac{N^{1-\alpha} \log N}{T^{1-\alpha}}} \right) \cdot \sqrt{C_6 N^\alpha T^{1-\alpha} \log NT}$$

$$= C_5 \sqrt{C_6} \sqrt{\frac{N^{1-\alpha} \log N}{T^{1-\alpha}} \cdot N^\alpha T^{1-\alpha} \log NT}$$

$$= C_5 \sqrt{C_6} \cdot \sqrt{N \log N \log NT}.$$

This shows that the product of the estimation error and the square root of regret scales as $\widetilde{\mathcal{O}}(\sqrt{N})$. This matches the lower bound established in Theorem 3.2 up to logarithmic factors. Therefore, Algorithm 5 is Pareto optimal for the trade-off between parameter estimation error and cumulative regret. $\square$

## F IMPLICATIONS FOR THE PAIRWISE DIFFERENCE ESTIMATION ERROR METRIC

We clarify here why our results, stated in terms of the estimation error $e_T(\widehat{\mathbf{v}}, \mathbf{v}) = \mathbb{E}\big[\|\widehat{\mathbf{v}} - \mathbf{v}\|_\infty\big]$, directly imply bounds of the same order for the estimation error $e_T(\widehat{\Delta}, \Delta) = \mathbb{E}\big[\|\widehat{\Delta} - \Delta\|_\infty\big]$ used in Zuo & Qin (2025), where $\Delta = [\Delta_{ij}]_{1 \leq i < j \leq N} = [v_i - v_j]_{1 \leq i < j \leq N}$ and $\widehat{\Delta}$ is an estimator of $\Delta$.

---

[5]While Theorem 1 in Miao & Chao (2021) is stated for Bayesian regret, it can be adapted to bound the frequentist regret with minor adjustments, as discussed in their Section 3.2.

**Upper bounds.** Given any estimator $\widehat{\mathbf{v}}$ of $\mathbf{v}$, consider the natural plug-in estimator $\widehat{\Delta}_{ij} = \widehat{v}_i - \widehat{v}_j$. By the triangle inequality, $|\widehat{\Delta}_{ij} - \Delta_{ij}| = |(\widehat{v}_i - \widehat{v}_j) - (v_i - v_j)| \leq |\widehat{v}_i - v_i| + |\widehat{v}_j - v_j|$. Taking the maximum over $1 \leq i < j \leq N$ and then expectation yields $e_T(\widehat{\Delta}, \Delta) = \mathbb{E}\big[\|\widehat{\Delta} - \Delta\|_\infty\big] \leq 2\mathbb{E}\big[\|\widehat{\mathbf{v}} - \mathbf{v}\|_\infty\big] = 2e_T(\widehat{\mathbf{v}}, \mathbf{v})$. Thus any upper bound on $e_T(\widehat{\mathbf{v}}, \mathbf{v})$ immediately implies an upper bound of the same order on $e_T(\widehat{\Delta}, \Delta)$.

**Lower bounds.** For the lower bounds, we consider an easier problem in which the learner is given perfect knowledge of $v_1$. Any minimax lower bound proven for this easier problem automatically applies to the original (harder) setting without this extra information.

Under this assumption, define $\widehat{\mathbf{v}}_{-1} = [\widehat{v}_i]_{2 \leq i \leq N}$ and $\mathbf{v}_{-1} = [v_i]_{2 \leq i \leq N}$. For $i \geq 2$, we have $\Delta_{1i} = v_1 - v_i$ and the corresponding estimator $\widehat{\Delta}_{1i} = v_1 - \widehat{v}_i$ (using the known $v_1$). Then $|\widehat{\Delta}_{1i} - \Delta_{1i}| = |(v_1 - \widehat{v}_i) - (v_1 - v_i)| = |\widehat{v}_i - v_i|$. Hence $\|\widehat{\Delta} - \Delta\|_\infty \geq \max_{2 \leq i \leq N} |\widehat{\Delta}_{1i} - \Delta_{1i}| = \max_{2 \leq i \leq N} |\widehat{v}_i - v_i| = \|\widehat{\mathbf{v}}_{-1} - \mathbf{v}_{-1}\|_\infty$, and therefore $e_T(\widehat{\Delta}, \Delta) \geq e_T(\widehat{\mathbf{v}}_{-1}, \mathbf{v}_{-1})$. Our lower-bound construction can then be applied directly to the $(N-1)$-dimensional vector $\mathbf{v}_{-1}$, yielding a minimax lower bound of the same order as in the $N$-dimensional case.

Putting these pieces together, our upper and lower bounds for $e_T(\widehat{\mathbf{v}}, \mathbf{v})\sqrt{\mathrm{Reg}(\pi, \mathbf{v})}$ imply upper and lower bounds of the same order for $e_T(\widehat{\Delta}, \Delta)\sqrt{\mathrm{Reg}(\pi, \mathbf{v})}$, which is the performance measure considered by Zuo & Qin (2025).

## G   AN ANYTIME VERSION OF THE TRADE-OFF ALGORITHM

In this section, we present an anytime variant of our trade-off algorithm for the $K$-capacitated MNL bandit problem (Algorithm 2). This version does not require prior knowledge of the time horizon $T$.

---

**Algorithm 6** Anytime Regret-Estimation Error Trade-off for $K$-Capacitated MNL Bandit

---

1: **Input:** number of products $N$, revenue vector $\mathbf{r} = (r_1, \ldots, r_N) \in [0,1]^N$, assortment capacity $K$, trade-off parameter $\alpha \in [0, 1/2]$, Algorithm 2, initial time block size $T_0$
2: Phase index $m = 0$, time step $t = 1$
3: **while** not terminated **do**
4:     Set phase length $L_m = 2^m T_0$
5:     Run Algorithm 2 with time horizon $L_m$ and trade-off parameter $\alpha$ for the next $L_m$ time steps, starting from time step $t$
6:     Update $t \leftarrow t + L_m, m \leftarrow m + 1$
7: **end while**
8: **Output:** sequence of assortments $S_1, S_2, \cdots \subseteq [N]$, attraction parameter estimates $\widehat{v}_1, \ldots, \widehat{v}_N$

---

In Algorithm 6, the learner employs a doubling-trick approach, dividing the time into phases of exponentially increasing lengths. In each phase, it runs Algorithm 2 with the specified trade-off parameter $\alpha$.

Algorithm 6 achieves the same Pareto optimal trade-off between estimation error and regret as Algorithm 2. To see this, consider any time horizon $T$. The regret upper bound follows the standard analysis of the doubling trick, which matches the regret bound of Algorithm 2. For the estimation error, by time $T$ the algorithm will have completed several full phases and will be partway through the next one. The total number of steps devoted to estimation is at least as large as the number of estimation steps in Algorithm 2. This ensure the same estimation error upper bound as in Theorem 4.1. Thus, Algorithm 6 maintains the same Pareto optimal trade-off guarantees without requiring prior knowledge of the time horizon.

## H   ADDITIONAL NUMERICAL RESULTS

In this section, we present numerical experiments comparing Algorithm 2 with the MNLExperimentUCB algorithm proposed by Zuo & Qin (2025). Because MNLExperimentUCB is defined only for

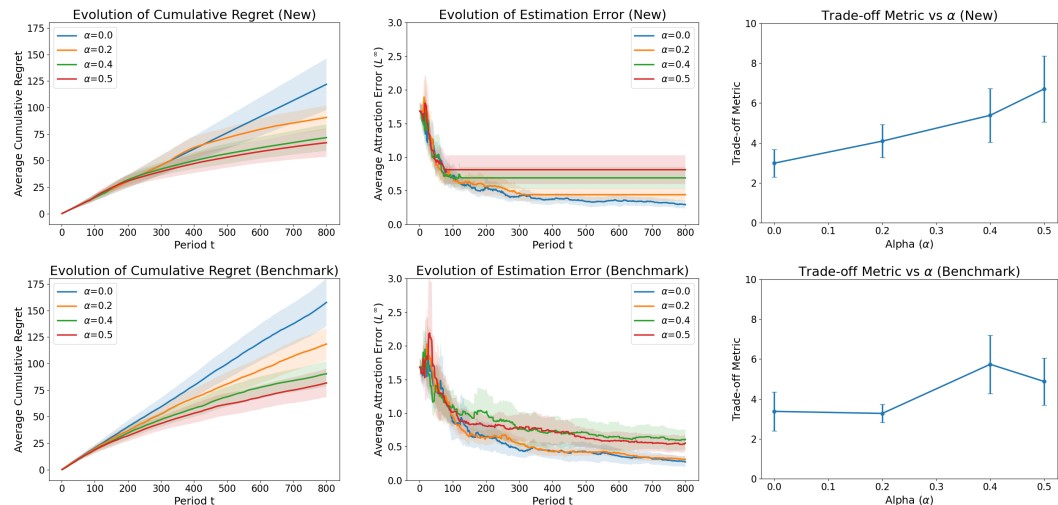

Figure 2: Comparison of estimation-regret trade-offs in the uncapacitated MNL bandit ($N$=8, $K$=8, $T$=800, $V$=2). Top: proposed Algorithm 2: (left) cumulative regret, (middle) estimation error over time, (right) trade-off metric vs. $\alpha$. Bottom: MNLExperimentUCB baseline with the same three panels. Curves are means over 20 runs with 95% confidence intervals.

the uncapacitated setting (i.e., without capacity constraints), all experiments here are conducted in that regime.

**Experimental setup.** In our simulations, we consider a setting with $N = 8$ products and capacity $K = 8$. The time horizon is $T = 800$. For each simulation run, the true attraction parameters $v_i$ are drawn uniformly from $[0, V]$ with $V = 2$, and the revenues $r_i$ are drawn uniformly from $[0, 1]$. We evaluate Algorithm 2 and the MNLExperimentUCB algorithm for trade-off parameters $\alpha \in \{0, 0.2, 0.4, 0.5\}$. The results are averaged over 20 independent runs.

**Results and analysis.** Figure 2 shows that our algorithm attains consistently lower cumulative regret than the MNLExperimentUCB baseline for all considered values of $\alpha$. The estimation errors are close when $\alpha$ is small; as $\alpha$ approaches $1/2$, our method exhibits a slightly higher error than the baseline. Consequently, the resulting trade-off metric remains close across the two algorithms over the examined range of $\alpha$.

These experiments demonstrate that Algorithm 2 attains performance comparable to the baseline of Zuo & Qin (2025) in the uncapacitated setting. In addition, Algorithm 2 naturally extends to capacitated instances, whereas the baseline is defined only for the uncapacitated case, highlighting the broader applicability of our approach.

