# OpenReview forum: "Optimal Trade-offs between Regret and Estimation in Capacitated Multinomial Logit Bandits"
_ICLR.cc/2026/Conference — Submitted to ICLR 2026_

### Official Review · Reviewer_Ga2Q · 2025-10-22

**Soundness:** 2
**Presentation:** 3
**Contribution:** 2
**Rating:** 4
**Confidence:** 4

**Summary:**

This paper investigates the trade-off between regret minimization and parameter estimation in Multinomial Logit (MNL) bandits.
Building on the work of Zuo & Qin (2025), the authors extend the analysis to the capacitated setting, where the assortment size is limited to
$K$. They establish tight lower and upper bounds on the trade-off between regret and customer attraction estimation error, and further propose an algorithm that achieves the optimal trade-off rate.
Additionally, the framework is extended to the joint assortment and pricing problem, demonstrating its broader applicability.

**Strengths:**

The paper provides the first tight characterization of the regret–estimation trade-off in capacitated MNL bandits, addressing an open problem left by Zuo & Qin (2025).
However, the comparison with prior work may be **misleading**, since the definition of estimation error used here differs from that in Zuo & Qin (2025).
While I agree with the authors that their definition—based on direct estimation of attraction parameters (as noted in Footnote 1)—is more appropriate than using relative pairwise differences, the claimed improvement over Zuo & Qin (2025) overlooks the fact that the two metrics correspond to different performance objectives, making the comparison not entirely fair.
In my view, the main contribution of this paper lies in redefining the estimation error in a more appropriate way and establishing the corresponding optimal trade-offs under this new formulation, rather than in claiming an improvement over the previous work.

**Weaknesses:**

**1. Technical concern in the proof of Theorem 3.2:**
In the instance construction, the revenue is set to 1 for one item and 0 for all others. Since the revenues are known to the algorithm, it can simply ignore the zero-revenue items and always select the item with revenue 1. Such a simple policy would then incur zero regret, which would make the lower bound on the trade-off measure equal to zero.
Could the authors clarify how this issue is resolved in your setup?
If I have misunderstood the setting, please correct me. I would be happy to raise my score once this point is addressed.

**2. Performance comparison with Zuo & Qin (2025):**
As mentioned above, the definition of estimation error used in this paper differs from that in Zuo & Qin (2025). Therefore, I believe that the authors’ claim of improving upon the previous results is somewhat overstated and potentially misleading, since the two formulations measure different performance objectives and are not directly comparable.


**3. Experiments:**
1) In Figure 1, only the regret and estimation errors are reported. However, it would be more informative to also include the trade-off metric.
Including this metric would make the results more convincing, especially if it demonstrates that the trade-off measure remains approximately independent of $\alpha$, consistent with the theoretical claims.

2) The confidence intervals are not shown in Figure 1.

3) No baseline algorithms are included for comparison. At a minimum, the authors could compare the regret performance of their proposed method with that of the algorithm from Zuo & Qin (2025).

**Questions:**

1. In Figure 1, the estimation error appears unchanged. However, unless I am missing something, in Algorithm 3 the parameter estimates are still being updated during the regret minimization phase.
Could the authors clarify why the estimation error remains constant even though the parameters are being updated?

2. What happens if $\alpha > 1/2$? Does the analysis break down in this regime? If so, please clarify which steps in the proofs.

---

> ### Author Response · Authors · 2025-11-20
> **Response (Part 1/2)**
>
> Thank you for your feedback and constructive suggestions! Please find our feedback below.
>
> **W1. Technical concern in the proof of Theorem 3.2.**
>
> Thank you for your careful reading and pointing this out! Your point is well taken, and we agree that under the policy you describe, our previous construction indeed yields zero regret. In the revised version, we have fixed the proof by constructing a new instance with $2N$ items (without loss of generality): the first $N$ items have revenue $1$, and the last $N$ items have revenue $0$. The concern you raise corresponds to policies that do not explore the zero-revenue items. In that case, for the attraction parameters of the first $N$ items we adopt the construction from Chen \& Wang (2018), which yields the regret lower bound $\mathrm{Reg}(\pi,\mathbf{v}) = \Omega(\sqrt{NT})$. Moreover, the estimation error is a constant $e\_T(\hat{\mathbf{v}},\mathbf{v}) = C\_E$, since the last $N$ parameters are not estimated. Combining this with the Assumption 2.1 of large horizon $T\gtrsim N$, the metric $e\_T(\hat{\mathbf{v}},\mathbf{v})\sqrt{\mathrm{Reg}(\pi,\mathbf{v})}$ is lower bounded by $\Omega(\sqrt{N})$. In the alternative case when the zero-revenue items are offered, the proof is similar to the original one and yields the same lower bound. We have updated the proof in the document; see the details in Appendix B.2.
>
>
> **W2. Performance comparison with Zuo \& Qin (2025) and difference on the definition of estimation error.**
>
> Thank you for this thoughtful comment. We clarify here why our results, stated in terms of the estimation error $e\_T(\hat{\mathbf{v}}, \mathbf{v}) = \mathbb{E}[\|\hat{\mathbf{v}} - \mathbf{v}\|\_\infty]$, directly imply bounds of the same order for the estimation error $e\_T(\hat{\Delta}, \Delta) = \mathbb{E}[||\hat{\Delta} - \Delta||\_\infty ]$ used in Zuo \& Qin (2025), where $\Delta = [\Delta\_{ij}]\_{1\leq i<j\leq N} = [v\_i - v\_j]_{1\leq i<j\leq N}$ and $\hat{\Delta}$ is an estimator of $\Delta$.
>
> **Upper bounds.**
>
> Given any estimator $\hat{\mathbf{v}}$ of $\mathbf{v}$, consider the natural plug-in estimator $\hat{\Delta}\_{ij} = \hat{v}\_i - \hat{v}\_j$. By the triangle inequality, $$
> |\hat{\Delta}\_{ij} - \Delta\_{ij}|
> = |(\hat{v}\_i - \hat{v}\_j) - (v\_i - v\_j)|
> \leq |\hat{v}\_i - v\_i| + |\hat{v}\_j - v\_j|.$$
>
> Taking the maximum over $1 \leq i < j \leq N$ and then expectation yields $
> e\_T(\hat{\Delta}, \Delta)
> = \mathbb{E}[||\hat{\Delta} - \Delta||\_\infty]
> \leq 2\mathbb{E}[||\hat{\mathbf{v}} - \mathbf{v}||\_\infty]
> = 2e_T(\hat{\mathbf{v}}, \mathbf{v})$. Thus any upper bound on $e\_T(\hat{\mathbf{v}}, \mathbf{v})$ immediately implies an upper bound of the same order on $e_T(\hat{\Delta}, \Delta)$.
>
> **Lower bounds.**
>
> For the lower bounds, we consider an easier problem in which the learner is given perfect knowledge of $v\_1$. Any minimax lower bound proven for this easier problem automatically applies to the original (harder) setting without this extra information.
>
> Under this assumption, define $\hat{\mathbf{v}}\_{-1} = [\hat{v}\_i]\_{2 \leq i \leq N}$ and $\mathbf{v}\_{-1} = [v\_i]\_{2 \leq i \leq N}$. For $i \geq 2$, we have $\Delta\_{1i} = v\_1 - v\_i$ and the corresponding estimator $\hat{\Delta}\_{1i} = v\_1 - \hat{v}\_i$ (using the known $v\_1$). Then $$
> |\hat{\Delta}\_{1i} - \Delta\_{1i}|
> = |(v\_1 - \hat{v}\_i) - (v\_1 - v\_i)|
> = |\hat{v}\_i - v\_i|.$$
> Hence
> $$
> ||\hat{\Delta} - \Delta||\_\infty
> \geq \max\_{2 \leq i \leq N} |\hat{\Delta}\_{1i} - \Delta\_{1i}|
> = \max\_{2 \leq i \leq N} |\hat{v}\_i - v\_i|
> = ||\hat{\mathbf{v}}\_{-1} - \mathbf{v}\_{-1}||\_\infty,$$
> and therefore $$
> e\_T(\hat{\Delta}, \Delta)
> \geq e\_T(\hat{\mathbf{v}}\_{-1}, \mathbf{v}\_{-1}).$$
> Our lower-bound construction can then be applied directly to the $(N-1)$-dimensional vector $\mathbf{v}\_{-1}$, yielding a minimax lower bound of the same order as in the $N$-dimensional case.
>
> Putting these pieces together, our upper and lower bounds for $e\_T(\hat{\mathbf{v}}, \mathbf{v}) \sqrt{\mathrm{Reg}(\pi, \mathbf{v})}$ imply upper and lower bounds of the same order for $e\_T(\hat{\Delta}, \Delta)\sqrt{\mathrm{Reg}(\pi, \mathbf{v})}$, which is the performance measure considered by Zuo \& Qin (2025). We have added this clarification to the revised manuscript; see Appendix F for details.

---

> ### Author Response · Authors · 2025-11-20
> **Response (Part 2/2)**
>
> **W3. Experiments.**
>
> Thank you for this comment. In Figure 1 of the revised manuscript, we have added the trade-off metric and the confidence intervals. We have also included a comparison with the algorithm of Zuo \& Qin (2025) in Appendix H; see details there.
>
> **Q1. In Figure 1, the estimation error appears unchanged. However, unless I am missing something, in Algorithm 3 the parameter estimates are still being updated during the regret minimization phase. Could the authors clarify why the estimation error remains constant even though the parameters are being updated?**
>
> It is true that the parameters can still be updated during the regret minimization phase in Algorithm 3. However, the estimation performed in the estimation phase is already sufficient to achieve the optimal trade-off. In the proof of Theorem 4.2, we use only the estimator obtained in the estimation phase to derive an upper bound for the trade-off metric that matches the lower bound in Theorem 3.2. Even if we continue to update the parameters during the regret minimization phase, the estimation error cannot improve beyond that achieved in the estimation phase. Therefore, in Algorithm 2 and in our experimental implementation, we freeze the parameter estimates after the estimation phase.
>
> **Q2. What happens if $\alpha>1/2$? Does the analysis break down in this regime? If so, please clarify which steps in the proofs.**
>
> If $\alpha > 1/2$, under Assumption 2.1 (large horizon $T\gtrsim N$), the regret upper bound $\mathrm{Reg}\_T(\pi, \mathbf{v}) = \tilde{\mathcal{O}}(N^{\alpha} T^{1-\alpha})$ stated in Theorem 4.2 would contradict the known lower bound $\Omega(\sqrt{NT})$ for MNL bandits (Chen \& Wang, 2018). Thus, our analysis does not extend to $\alpha > 1/2$. The failure occurs in the proof of Theorem 4.2, Part (ii): when $\alpha > 1/2$, the term $N^{\alpha} T^{1-\alpha}$ is no longer dominant; instead, $\sqrt{NT}$ governs the regret, yielding a suboptimal trade-off metric. An analogous limitation also appears in Zuo \& Qin (2025) for $\alpha > 1/2$.
>
> We also comment on the boundary cases $\alpha = 0$ and $\alpha = 1/2$. Notice that when $\alpha = 0$, the estimation-error upper bound $\tilde{\mathcal{O}}(\sqrt{N/T})$ in Theorem 4.2 matches the lower bound in Theorem 3.1, showing that the estimation error is optimal. When $\alpha = 1/2$, the regret upper bound $\tilde{\mathcal{O}}(\sqrt{NT})$ in Theorem 4.2 matches the lower bound of Chen \& Wang (2018), showing that the regret is optimal. In both cases, the trade-off metric is optimal. Hence $\alpha \in [0, 1/2]$ is the critical range in the analysis.

---

> > ### Comment · Reviewer_Ga2Q · 2025-11-27
> >
> > Thank you for the detailed responses. My concerns have been largely addressed; thus, I will revise my score accordingly.

---

### Official Review · Reviewer_HS3S · 2025-10-29

**Soundness:** 3
**Presentation:** 3
**Contribution:** 3
**Rating:** 4
**Confidence:** 3

**Summary:**

This paper investigates the fundamental trade-off between parameter estimation and regret minimization in the context of capacitated Multinomial Logit bandits. The paper's primary contribution is a tight characterization of this trade-off. The authors first establish a minimax lower bound, proving that for any algorithm, the product of the estimation error and the square root of the regret must be at least $\Omega(\sqrt{N})$. They then propose an algorithm that can be tuned by a parameter $\alpha \in [0, 1/2]$. This algorithm is proven to achieve a matching upper bound of $\widetilde{\mathcal{O}}(\sqrt{N})$ for the same trade-off metric, making it Pareto-optimal.

**Strengths:**

1. The paper resolves the dependency on $N$, which was left open by prior work (Zuo & Qin, 2025). The $\Omega(\sqrt{N})$ lower bound and matching algorithm significantly improve upon the previous $\Omega(1)$ and $\widetilde{\mathcal{O}}(N^{3/4})$ bounds for the uncapacitated case.
2. The paper is exceptionally well-written. The use of a table (Table 1) to compare related work is very helpful.

**Weaknesses:**

1. The model assumes product revenues $r_i$ are known and fixed. In many real-world scenarios, the revenue or margin might be part of the optimization problem or uncertain. A discussion of how this assumption impacts the results or how it could be relaxed in future work would be valuable.
2. The notation in Step 14 of Algorithm is confusion.
3. The argument for Theorem 3.3 currently relies on asymptotic arguments (i.e., big-O notation). A more rigorous proof should make the argument non-asymptotic by providing explicit constants to substantiate the claimed sufficient condition for Pareto optimality.
4. Definition 2.1 lacks sufficient precision. It is unusual to define a comparison between two quantities based on an imprecise metric. For example, consider $A=(1,2)$ and $B=(2,1)$. Depending on Definition 2.1, one could claim that either $A$ Pareto dominates $B$ or $B$ Pareto dominates $A$, which illustrates the ambiguity in the current definition.

**Questions:**

see the weaknesses.

---

> ### Author Response · Authors · 2025-11-20
>
> Thank you for your feedback and constructive suggestions! Please find our feedback below.
>
> **W1: The model assumes product revenues $r\_i$ are known and fixed. In many real-world scenarios, the revenue or margin might be part of the optimization problem or uncertain. A discussion of how this assumption impacts the results or how it could be relaxed in future work would be valuable.**
>
> Thank you for this comment. Section 5.2 presents an extension to the joint assortment and pricing problem, which relaxes the fixed-revenue assumption. In this setting, the effective revenue of item $i$ in period $t$ is $r\_i + p\_{ti}$, where $p\_{ti}$ denotes the pricing decision variable in the optimization problem. We show that our regret-estimation trade-off results extend to this joint assortment and pricing setting. For more general uncertain revenue settings, it remains an interesting open question for future research to characterize the impact of revenue uncertainty on the regret-estimation trade-off.
>
>
> **W2. The notation in Step 14 of Algorithm is confusion.**
>
> Thank you for pointing this out. We have revised the notation in Step 14 of Algorithm 1 for better clarity. Please see the revised Algorithm 1 in the manuscript for the changes.
>
>
>
> **W3, W4. Both the Definition 2.1 and the argument for Theorem 3.3 currently relies on asymptotic notations (i.e., big O notation).**
>
> Thank you for this comment. Our notion of Pareto optimality focuses on the *rates* in $T$ and $N$ rather than on tight constants, which is consistent with prior work (Simchi-Levi \& Wang, 2025; Zuo \& Qin, 2025). Note that Pareto dominance is a partial order, not a total order. In your example, because both $A$ and $B$ have the same rates (constant) in $T$ and $N$, neither $A$ Pareto dominates $B$ nor $B$ Pareto dominates $A$; this is consistent with the definition.
>
> We admit that there are many artificial terms (all absolute constants) introduced in our proofs of both the lower and upper bounds. This prevents us from explicitly writing down the constant in Theorem 3.3. On the other hand, we emphasize that by big O notation we mean dominance up to *absolute constants*, so our result holds in a non-asymptotic sense as long as $T \gtrsim N$, as assumed in Assumption 2.1.

---

> ### Comment · Reviewer_HS3S · 2025-11-25
>
> We maintain our original evaluation. Under your current definition, which ignores absolute constants, one could construct cases (e.g., $A = [3,2]$) where another vetor $B=[1,1]$ could also “dominate” $A$, which is not reasonable.Meanwhile, relying on asymptotic notations in the proofs is not rigorous.

---

### Official Review · Reviewer_4TxG · 2025-10-31

**Soundness:** 3
**Presentation:** 3
**Contribution:** 3
**Rating:** 4
**Confidence:** 4

**Summary:**

This paper investigates online MNL bandits with two objectives: minimizing cumulative regret and estimating the attraction vector $v$. The first contribution is an information theoretic lower bound: to attain estimation error $e_T(v,\hat v)\le \varepsilon$, one needs on the order of $N/\varepsilon^{2}$ samples. The authors then prove a fundamental trade-off
$
e_T(v,\hat v)\sqrt{\mathrm{Reg}_T}\ge \tilde{\Omega}(\sqrt{N}),
$
which formalizes that both goals cannot be achieved simultaneously beyond this rate. A pure-exploration procedure meets this optimal product but suffers linear regret in $T$. To move along the Pareto frontier, the paper proposes an explore-then-exploit algorithm that delivers sublinear regret while controlling $e_T$, though with a worse dependence on the value scale $V$. Experiments support the theory and illustrate the regret–estimation trade-off.

**Strengths:**

- The paper provides a clear and formal treatment of the trade-off between regret minimization and value estimation in online MNL bandits.
 - The lower bound $e_T(v,\hat v)\sqrt{\mathrm{Reg}_T}\geq\tilde{\Omega}(\sqrt{N})$ is interesting in MNL that establishes the Pareto frontier between learning and exploration.
- The theoretical results are supported by experiments that effectively demonstrate the empirical trade-off.
 - I read the proofs in general and it looks correct to me based on the parts I have checked.

**Weaknesses:**

- The motivation for focusing on *value* estimation (estimating $v$) is not entirely clear. In most practical MNL applications, the primary objective is to maximize expected revenue rather than the latent value vector $v$ itself. Since the ultimate performance metric in these problems is revenue-based, it is unclear why directly estimating $v$ becomes a natural or meaningful intermediate goal.
- The algorithmic contributions are relatively straightforward. The method achieving the optimal trade-off is essentially a *pure-exploration* algorithm, while the one balancing estimation error and regret is a classic *explore-then-learning* procedure that simply switches to a known MNL-UCB algorithm from prior work. There is limited algorithmic innovation or adaptivity: the trade-off parameter $\alpha$ is also preset rather than learned. Consequently, while the theoretical analysis is neat, the algorithmic side feels largely incremental.
- The dependence on the value scale $V$ appears suboptimal and is not theoretically justified. For example, the pure-exploration algorithm exhibits a $V^{3/2}$ dependence, while the explore-then-learning algorithm incurs a $V^2$ factor. Since the lower bounds presented in the paper do not contain any explicit $V$ term, it remains unclear whether these dependencies are information-theoretically necessary or simply artifacts of the analysis. This weakens the optimality claim of the trade-off frontier, especially in practical settings where $V$ can be very large (like contextual case where $v_i=\exp(\langle x,\theta_i\rangle)$).

**Questions:**

- 1. Can the authors clarify the motivation behind focusing on *value* estimation rather than directly estimating choice probabilities or maximizing expected revenue? In what types of downstream applications would accurate recovery of $v$ itself be the key goal?
- 2. The current analysis exhibits an explicit dependence on the value scale $V$ in the upper bound, yet the lower bound does not include $V$. Can the authors provide intuition or formal arguments suggesting whether this dependence is fundamental? In practical or contextual MNL settings where $v_i = \exp(\langle x, \theta_i \rangle)$ and $V$ can be exponentially large, so characterizing the dependency on $V$ can be important in practice.
- 3. The algorithms considered are largely non-adaptive. Do the authors expect that an online, single-phase approach (e.g., optimism-based or posterior-sampling) could achieve a similar Pareto frontier without a hard phase separation between exploration and exploitation?

**Details Of Ethics Concerns:**

None.

---

> ### Author Response · Authors · 2025-11-20
>
> Thank you for your feedback and constructive suggestions! Please find our feedback below.
>
> **Q1: Can the authors clarify the motivation behind focusing on value estimation rather than directly estimating choice probabilities or maximizing expected revenue? In what types of downstream applications would accurate recovery of $v$ itself be the key goal?**
>
> Thank you for this question. Choice probabilities and expected revenue are both functions of the attraction parameters $\mathbf{v}$. Accurately estimating $\mathbf{v}$ enables accurate estimation of these derived quantities. Moreover, in many practical applications the attraction parameters themselves are of direct interest. For example, in marketing and consumer behavior analysis they offer insight into customer preferences and product attractiveness.
>
> In Section 5, we discussed two direct applications of accurate estimation of $\mathbf{v}$: (i) customer preference estimation, where the same guarantees continue to hold; and (ii) the joint assortment and pricing problem, which yields new insights into the regret-estimation trade-off in broader contexts.
>
>
>
> **Q2. The current analysis exhibits an explicit dependence on the value scale $V$ in the upper bound, yet the lower bound does not include $V$. Can the authors provide intuition or formal arguments suggesting whether this dependence is fundamental? In practical or contextual MNL settings where $v\_i = \exp{\langle x, \theta\_i\rangle}$ and $V$ can be exponentially large, so characterizing the dependency on $V$ can be important in practice.**
>
>
> Thank you for this insightful question. We note that the dependence on the value scale $V$ is fundamental, and Theorem 3.1 can be directly generalized to include this dependence. If we change the parameter instance into $\mathbf{v} = (v\_1,\dots, v\_N) = (V, \dots, V)$ for some constant $V>1$, and define $\mathbf{v}'$ accordingly, the same proof technique still applies. In this case, we can show that
> $$
> \text{D}\_{\text{KL}}(\mathbb{P}\_{\mathbf{v}}(\cdot\mid S\_t)\|\mathbb{P}\_{\mathbf{v}'}(\cdot\mid S_t)) \leq \frac{2\varepsilon^2}{V(1 + V|S\_t|)}\mathbf{1}\\{j\in S\_t\\}.
> $$
> As a result, Theorem 3.1 can be generalized to
> $$
> \inf\_{(\pi,\hat{\mathbf{v}})}\sup\_{(\mathbf{v}, \mathbf{r})\in\mathcal{E}}e\_T(\hat{\mathbf{v}}, \mathbf{v}) \geq \frac{V}{16}\sqrt{\frac{N}{T}}.
> $$
> This yields a lower bound with explicit dependence on the scale $V$ of the attraction parameters. As a direct result, the trade-off metric $e\_T(\hat{\mathbf{v}}, \mathbf{v})\sqrt{\mathrm{Reg}(\pi, \mathbf{v})}$ is lower bounded by $\Omega(V\sqrt{N})$.
>
> However, we acknowledge that the dependence on $V$ in the upper bounds of Theorems 4.1 and 4.2 is suboptimal. Closing this gap remains an interesting open question for future research. We have added this discussion in the revised manuscript; see Remark B.1 for details.
>
>
>
>
> **Q3: The algorithms considered are largely non-adaptive. Do the authors expect that an online, single-phase approach (e.g., optimism-based or posterior-sampling) could achieve a similar Pareto frontier without a hard phase separation between exploration and exploitation?**
>
> Thank you for this thoughtful comment. We add an anytime algorithm using the doubling-trick. This algorithm does not require the prior knowledge of the time horizon $T$ and alternates between exploration and exploitation in each epoch. This anytime algorithm achieves the same Pareto frontier as our two-phase algorithm. We have added a discussion of this in the revised manuscript; please see the details in Appendix G.

---

> ### Comment · Reviewer_4TxG · 2025-11-27
>
> Thanks the authors for their detailed responses to my questions. However, I am still not convinced in the motivation of this problem since in both of the applications that is referred in Section 5, the main objective is still the revenue or the reward instead of the true value that induces this revenue/reward. It is not clear to me why the balance/trade-off between these two should be considered in practice. From the technical perspective, I agree that using a standard doubling-trick can bypass the knowledge of $T$, however, I feel that technically, the algorithm is mostly applying standard analysis and may lack the technical contribution. Therefore, I tend to maintain my current score.

---

### Official Review · Reviewer_3nSy · 2025-11-01

**Soundness:** 4
**Presentation:** 3
**Contribution:** 3
**Rating:** 8
**Confidence:** 4

**Summary:**

The paper studies the Pareto trade-off between regret minimization and parameter estimation in capacitated MNL bandits. To this end, the paper shows a lower bound on $e_T(\hat{v},v).R(T)$ (Theorem 3.2). They also point out a mistake in older, similar proofs and correct it in their presentation. From an algorithm design perspective, they present a two-phase algorithm (Algorithm 2) that achieves this lower bound, utilizing a tunable parameter. Overall, the presentation is neat and contributions are novel, and in some way, they close a "gap" in the joint-regret and parameter estimation line of works for the MNL bandit.

**Strengths:**

1)Their theory results are state of the art, e.g., dependence w.r.t $N$. They also correct a "mistake" in the previous literature
2) The main text is largely well-written and explained well.
3)Their lower bound hardness construction, Theorem 3.1 proof, could be of independent interest.
4)While other aspects of the paper: two-phase algorithm, alpha parameter tuning appear standard and reasonable

**Weaknesses:**

1) Line 7 in Algorithm 3 is computationally heavy. The authors should discuss this or point out if there is an efficient way to implement Line 7
2)Assumption 4.2 (large horizon) is very strong. This may not be valid at all for a variety of problems.

I do not have much complaints with the paper.

**Questions:**

1) In Table 1, the revenue-based lower bound is $\Omega(1)$. can this be improved?
2)have the authors thought about anytime algorithm instead of a two-phase algorithm? Can a doubling-trick be used here?

Please also look into the other weaknesses as above.

---

> ### Author Response · Authors · 2025-11-20
>
> Thank you for your positive feedback and constructive suggestions! Please find our feedback below.
>
> **W1. Line 7 in Algorithm 3 is computationally heavy. The authors should discuss this or point out if there is an efficient way to implement Line 7.**
>
> Thanks for pointing this out. There are efficient polynomial time algorithms to solve the static assortment optimization problem in Line 7 under MNL model with known parameters (see [1], [2], [3]). We have added this clarification in the revised manuscript; please see the details in Appendix C.3.
>
>
> **W2. Assumption 4.2 (large horizon) is very strong. This may not be valid at all for a variety of problems.**
>
> Thank you for this comment. First, we want to make an update on Assumption 4.2(now moved to Assumption 2.1 in revised version): We no longer need the condition `T sufficiently large', thus the only requirement on $T,N$ here is $T\gtrsim N$. This $T\gtrsim N$ condition is necessary for both meaningful estimation and regret minimization, given the $\Omega(\sqrt{N/T})$ estimation lower bound of estimation in Theorem 3.1 and the $\Omega(\sqrt{NT})$ regret lower bound shown in [4]
>
> **Q1. In Table 1, the revenue-based lower bound is $\Omega(1)$. Can this be improved?**
>
> Thank you for this insightful question. For single-item revenue estimation, if we additionally assume that all revenue parameters $r_i$ are bounded below by a small constant $\delta$, we can improve the revenue-based trade-off lower bound to $\Omega\big(\delta \sqrt{N}/(1+V)^2\big)$. Let $R_i = r_iv_i/(1+v_i)$ be the expected revenue of items $i$, and $\hat{R}\_i$ be its estimator. It suffices to show that $\min_{(\pi, \hat{\mathbf{v}})} \max_{(\mathbf{v}, \mathbf{r}) \in \mathcal{E}} e_T(\hat{\mathbf{v}}, \mathbf{v}) \ge \varepsilon$ implies $\min\_{(\pi, \hat{R}\_i)} \max\_{(\mathbf{v}, \mathbf{r}) \in \mathcal{E}} e_T(\hat{R}_i, R_i) \ge \delta \varepsilon /(1+V)^2$. The derivation is similar to that of Lemma D.1 using the contrapositive of this statement. For revenue estimation beyond the single-item case and without a positive lower bound on $r\_i$, it remains an open question whether the revenue-based trade-off lower bound can be improved beyond $\Omega(1)$.
>
> **Q2. Have the authors thought about anytime algorithm instead of a two-phase algorithm? Can a doubling-trick be used here?**
>
> Thank you for this thoughtful comment! **YES**, a doubling-trick can be used to convert our two-phase algorithm into an anytime algorithm. This algorithm does not require the prior knowledge of the time horizon $T$ and alternates between exploration and exploitation in each epoch. This anytime algorithm achieves the same Pareto frontier as our two-phase algorithm. We have added a discussion of this in the revised manuscript; please see the details in Appendix G.
>
>
>
>
>
>
>
> **References:**
>
> [1] Paat Rusmevichientong, Zuo-Jun Max Shen, and David B Shmoys. Dynamic assortment optimization with a multinomial logit choice model and capacity constraint. Operations research, 58(6): 1666–1680, 2010b.
>
> [2] James Davis, Guillermo Gallego, and Huseyin Topaloglu. Assortment planning under the multinomial logit model with totally unimodular constraint structures. Work in Progress, 2013
>
> [3] Vashist Avadhanula, Jalaj Bhandari, Vineet Goyal, and Assaf Zeevi. On the tightness of an lp relaxation for rational optimization and its applications. Operations Research Letters, 44(5):612–617, 2016
>
>
> [4] Xi Chen and Yining Wang. A note on a tight lower bound for capacitated mnl-bandit assortment selection models. Operations Research Letters, 46(5):534–537, 2018.

---

> > ### Comment · Reviewer_3nSy · 2025-11-25
> > **Thanks for the rebuttal**
> >
> > Thanks for carefully responding to my questions and concerns. Most of my queries were answered.
> >
> > I understand your submission was updated during the rebuttal period. It would be helpful for subsequent evaluation if the authors could summarize the major modifications they made during the rebuttal period.

---

> > > ### Author Response · Authors · 2025-11-27
> > >
> > > Thank you for your positive feedback! We added a general block to summarize the major adjustments made in the revised manuscript.

---

### Author Response · Authors · 2025-11-27
**Major adjustments in the revised manuscript**

We made the following major adjustments in the revised manuscript:

1. We added an anytime algorithm using the doubling-trick, which achieves the same Pareto frontier as our two-phase algorithm (see Appendix G).

2. We updated the proof of Theorem 3.2 to address the concern raised by Reviewer Ga2Q (see Appendix B.2 for details).

3. We clarified the relationship between our estimation error and the estimation error used in Zuo \& Qin (2025). Our results imply bounds of the same order for the estimation error used there (see Appendix F for details).

4. We added a discussion of the dependence on the value scale $V$ in the lower bounds of Theorems 3.1 and 3.2, which yields a lower bound of $\Omega(V\sqrt{N})$ for the trade-off metric (see Remark B.1 for details).

5. We added an experimental comparison with the algorithm of Zuo \& Qin (2025) (see Appendix H).

---

### Meta-Review · Area_Chair_4q8k · 2025-12-11

**Summary:**

The strengths of this paper includes that:

i) Clear theoretical development with tight bounds.

ii) Clear writing.

The main concern of this paper is that:

i) The proposed algorithms are straightforward, and the results appear incremental with limited novelty in terms of techniques..

Regarding the other concerns raised by reviewers, after carefully reading the paper, I believe they stem from some misunderstandings of the content rather than fundamental flaws in the work.

**Reviewer Concerns:**

I believe the misunderstandings raised by some reviewers are explained in a convincing manner and can be considered addressed. However, the main concern regarding the novelty of the work remains unresolved. After reading the paper myself, I also find that the proposed algorithms are quite straightforward and introduce very few new techniques. The rebuttal does not successfully articulate the novelty or the technical challenges of the approach, and I do not expect that the reviewers will be convinced on this point either.

**Reviewer Scores:**

Reviewer 3nSy and 4TxG will not change their scores.

Although reviewer HS3S claimed that he will not change his score, I believe this is still due to misunderstandings that could be clarified during the discussion phase. Therefore, it is possible that his score may increase by 1 point.

For reviewer Ga2Q, all of his concerns appear to be satisfactorily addressed, and I expect that his score will increase by 1.

---

### Decision · Program_Chairs · 2026-01-26

Reject